## RESEARCH ARTICLE

# Cell cycle dynamics regulate H3K27 and H3K9 histone modifications in *Drosophila*

Liyne Nogay[1,2], Ananthakrishnan Vijayakumar Maya[1,2‡], Lara Heckmann[1,2‡], Francesco Cardamone[1,2,3], Isabelle Grass[1,4], Aakriti Singh[1,2], Anna Frey[1,5], Laurin Ernst[1,2], Nicola Iovino[3,4], Anne-Kathrin Classen[1,4]*

1 Faculty of Biology, University of Freiburg, Freiburg, Germany, 2 International Max Planck Research School for Epigenetics, Biophysics and Metabolism, Freiburg, Germany, 3 Max Planck Institute of Immunobiology and Epigenetics, Freiburg, Germany, 4 CIBSS Centre for Integrative Biological Signalling Studies, University of Freiburg, Freiburg, Germany, 5 Spemann Graduate School of Biology and Medicine (SGBM), University of Freiburg, Freiburg, Germany

‡ Shared coauthors.
* anne.classen@biologie.uni-freiburg.de

## Abstract

Cell cycle progression presents a fundamental challenge to epigenome integrity, particularly due to the need to reestablish post-translational histone modifications (PTMs) following DNA replication. Although proliferative and differentiating tissues exhibit markedly different cell cycle dynamics, how these differences shape the histone modification landscape in vivo remains largely unexplored. Here, we show that levels of H3K27ac, H3K27me3, and H3K9me3 are tightly linked to cell cycle dynamics in the *Drosophila* wing imaginal disc. We demonstrate that both physiological and pathological elongation of the cell cycle led to an accumulation of H3K9me3 and H3K27me3, whereas cell cycle acceleration reduces their levels. In contrast, H3K27ac exhibits the opposite pattern: levels decrease in arrested cells and increase with faster cycling. Genome-wide CUT&Tag analysis reveals that these changes predominantly affect genomic loci already modified in normally proliferating tissue. Importantly, the regulation of methylation levels at H3K9 and H3K27 is not solely mediated by the cell cycle machinery but reflects a metabolically guided process in which the rate of methylation is coupled to the rate of cell proliferation through metabolic activity, including signaling via the Insulin/PI3K/Akt pathway. Our study thus reveals key principles for understanding histone methylation in proliferating, senescent, and differentiating cells. In contrast, H3K27 acetylation is regulated through a distinct, cell cycle-coupled mechanism. We find that CBP/Nejire-mediated acetylation of H3K27 peaks during early and late S-phase and is reversed by HDAC1, as cells exit replication. Together, our findings establish a robust link between cell cycle progression and histone modification dynamics, highlighting the necessity of maintaining balanced PTM levels under varying proliferative states. These insights have broad implications for our understanding of development, aging, and tumor growth.

**Data availability statement:** All data, workflows, and FIJI based algorithms necessary to interpret the imaging data are included within the manuscript. The points extracted from images for analysis, the values used to build graphs, the values behind the means, and statistics reported can be found in S1 File (Source data and statistics). The code used for the mapping, normalization, peak calling, and signal quantification is available on Github: https://github.com/LaraH9/nogay_et_al_2025. A version of record was uploaded to Zenodo: https://doi.org/10.5281/zenodo.18873410. The CUT&Tag sequencing data generated in this study are provided on NCBI SRA: https://www.ncbi.nlm.nih.gov/sra/PRJNA1300380.

**Funding:** Funding for this work was provided by the Deutsche Forschungsgemeinschaft (DFG, German Research Foundation) under Germany's Excellence Strategy (CIBSS – EXC-2189), the DFG Heisenberg Program to AKC (668189), as well as DFG grants to AKC (667603) and the Boehringer Ingelheim Foundation (BIF Plus3 & Rise Up) to AKC. The funders had no role in study design, data collection and analysis, decision to publish, or preparation of the manuscript.

**Competing interests:** The authors have declared that no competing interests exist.

**Abbreviations:** AED, after egg deposition; BDSC, Bloomington Drosophila Stock Center; CDKs, Cyclin-dependent kinases; DSHB, Developmental Studies Hybridoma Bank; HAT, histone acetyltransferase; HMH, Hilde Mangold House; LIC, Life Imaging Center; PTMs, post-translational histone modifications; Rb, retinoblastoma; ROIs, regions of interest; TSS, transcription start sites; ZNC, zone of nonproliferation cells.

## Introduction

The regulation of the cell cycle and chromatin needs to be tightly coordinated to preserve a functional epigenome. Specifically, progression through S-phase poses several challenges: chromatin must become transiently accessible for DNA replication, pre-existing post-translationally modified histones must be evenly distributed between daughter genomes and newly incorporated histones must acquire appropriate modifications to compensate for the semi-conservative dilution of modifications. These demands are complicated by the fact that cell cycle progression and dynamics differ drastically across biological contexts, ranging from rapid divisions in embryonic and regenerating tissues to complete arrest in terminally differentiated cells. How chromatin states are maintained across diverse proliferative tissue environments in vivo is still insufficiently understood.

In actively proliferating tissues, entry into the cell cycle at the G1/S transition, and progression through subsequent cycle phases, are tightly regulated. This regulation is mediated by Cyclins and Cyclin-dependent kinases (CDKs), as well as retinoblastoma (Rb) proteins and E2F transcription factors. Throughout the cell cycle, specific checkpoints respond to growth factors or cell size to modulate the rate of proliferation, as well as to DNA damage or chromosome misalignment to maintain the integrity of the genome [1–3]. As cells enter stages of differentiation, quiescence, or senescence, they typically withdraw from active cycling by either prolonging their G1 phase or entering a stable cell cycle arrest known as G0. This transition away from proliferation often involves CDK inhibitors, such as p21 and p27, which are intricately linked to gene expression programs that guide cell fate determination and differentiation [4–7]. Progression through the cell cycle presents significant challenges for chromatin organization and epigenetic regulation, especially during S-phase but also during mitosis. Specifically, during S-phase, chromatin must become highly accessible to facilitate replication fork progression, nucleosome redistribution and de novo nucleosome incorporation in the replicated genomes [7–12]. In contrast, in mitosis, chromatin must be tightly compacted to ensure accurate chromosome segregation [12,13]. Central to the regulation of chromatin organization are post-translational histone modifications (PTMs), which are also epigenetic regulators of transcriptional silencing and gene activation [14–16]. It is well-established that PTM dysregulation can lead to cell cycle defects by disrupting chromatin organization and entire gene regulatory networks [17–20]. While much attention has been dedicated to understanding how PTM dysregulation can drive pathogenesis by altering the expression of genes central to cell cycle regulation, less is known about how cell cycle dynamics connect to the maintenance and reestablishment of histone modifications. Yet, studies of the cell cycle-dependent regulation of the histone-modifying enzymes underscore the capacity of the cell cycle machinery to control specific histone modifications and thereby maintain chromatin function [21–23].

Some of the most abundant PTMs that regulate chromatin accessibility and transcriptional activity are acetylation and methylation of lysine residues on histone tails. While several lysine residues in histones can be acetylated or methylated, research has most strongly focused on dynamic acetylation and methylation of lysines 9 and

27 on histone H3 (H3K9 and H3K27), which represent some of the functionally most important histone modifications. In most species, trimethylation of H3K27 is mediated by Polycomb Repressive Complexes and relies on histone methyl-transferases of the Enhancer of zeste E(z) family. H3K27me3 maintains the silencing of many genes required for cell fate specification during development, most famously Hox genes [24–28]. In contrast, trimethylation of H3K9 (H3K9me3) is mediated by the enzymatic activity of Su(var)3-9 and is central to the formation and maintenance of constitutive heterochromatin at telomeres, centromeres, and repetitive repeats [29–31]. In *Drosophila*, H3K27 acetylation is mediated by the histone acetyltransferase (HAT) Nejire (Nej), a homolog of mammalian p300/CBP, and is reversed by the histone deacetylase HDAC1/Rpd3 [32–35]. H3K27ac is strongly associated with active promoters and enhancers, where it is thought to facilitate chromatin opening and transcriptional activation [36–40]. As a consequence of these important functions, dysregulation of H3K9me3, H3K27me3, and H3K27ac is a common feature in cancer or other diseases characterized by alterations and defects in cell proliferation. Yet, importantly, it often remains unclear if problems with epigenetic modifications cause proliferation defects or if vice versa proliferation defects cause problems with epigenetic modifications [17–19,41]. Disentangling this relationship is key to understanding how epigenetic and proliferative states influence one another in development and disease.

Importantly, the semi-conservative nature of DNA replication during S-phase has a profound impact on the landscape of these histone modifications. As the DNA is duplicated, pre-existing histones bearing PTMs are equally distributed onto the newly synthesized strands. Naïve, newly synthesized histones are integrated to ensure proper packaging of the duplicated genomes but also necessitating the reestablishment of the now diluted PTM code [11 42–44]. To reestablish the epigenetic histone modification landscape after incorporation of newly synthesized, naive histones, histone-modifying enzymes act post-replication by using the modifications on old recycled histones as a template to modify the newly incorporated histones [8,45,46]. Importantly, in cultured cells, histone acetylation marks are typically restored immediately after replication often via a transcription-dependent process, while histone methylation is generally reestablished during subsequent gap phases [9,10,47,48].

Of note, strong acetylation of newly synthesized histones plays a specific cell cycle-dependent role during S-phase. Hyperacetylation, for example, at H4K5, H4K12, H3K14, H3K23, or H3K56, is observed during S-phase and facilitates recognition of newly synthesized histones by histone chaperone complexes, promoting their correct incorporation into newly synthesized chromatin. Hyperacetylation of histones presumably occurs in the cytoplasm and may be mediated by B-type acetyltransferases like Hat1 [49–59].

Despite these insights, questions remain about how cell cycle progression influences histone modification dynamics, especially in in vivo settings. For instance, to what extent are histone modifications regulated in a cell cycle-dependent manner in developing tissues? How does the maintenance of these modifications differ between rapidly proliferating cells and cells in quiescent, senescent, or post-mitotic states? Moreover, how is the activity of histone-modifying enzymes coordinated with cell cycle checkpoints? To begin to address these questions we used the developing *Drosophila* wing imaginal disc to characterize cell cycle-dependent histone modifications and reveal pronounced dynamics of H3K9 and H3K27 modifications in a developing tissue.

## Results

### Post-translational modifications of H3K27 and H3K9 are linked to cell cycle dynamics

To understand how cell cycle dynamics may affect chromatin dynamics, we aimed to identify post-translational histone modifications that are sensitive to changes in cell cycle progression. Using the *rn-GAL4* driver (S1A Fig), under the control of a temperature-sensitive GAL80ts, we genetically manipulated cell cycle progression in the pouch of developing wing imaginal discs for 24 h by accelerating or decelerating cell cycle dynamics. Within this timeframe of 24 h, we expect to disturb at least one cell cycle, which lasts between 12 and 18 h in third instar wing imaginal discs [60–63]. First, we induced a cell cycle arrest in late G2 by expressing *Cdk1-RNAi* [64]. Second, ectopic expression of the ERK-responsive

ETS transcription factor Pointed-P1 (PntP1) causes cells to arrest in either G1 or in G2 [65]. We confirmed the arrest in our genotypes by observing the cell cycle reporter FUCCI and the absence of EdU incorporation, a nucleotide analogue visualizing DNA replication, in nuclei of the *rn-GAL4* expression domain (S1B–S1N Fig) [66]. Conversely, we accelerated the cell cycle by coexpressing of dE2F1 and dDP [60,67]. This resulted in elevated EdU incorporation and increased the proportion of cells in S-phase and decreased the proportion of cells in gap phases, reflecting the accelerated S-phase and cell cycle dynamics of this genotype (S2A–S2FFig).

As post-translational histone modifications are numerous, we focused on prominent modifications mediated by well-known histone-modifying complexes. We analyzed modifications associated with gene regulation, such as H3K27ac (mediated by the p300/CBP Nejire), H3K9ac (mediated by SAGA-HAT complex), H3K4me3 (mediated by the Trithorax (Trx)) and H3K27me3 (mediated by the Polycomb protein E(z)), or modifications associated with genome maintenance, such as H3K9me3 (mediated by Su(var)3-9), as well as examples of less well characterized modifications, such as H3K18ac, H4K8ac and H3K18 crotonylation (mediated by p300/CBP Nejire), and pan-H3K9 methylation or total lysine acetylation [15,32,68,69]. We performed immunofluorescence staining for these modifications in wing imaginal discs with the induced cell cycle alterations described above.

Many of the histone modifications analyzed did not change upon manipulating cell cycle dynamics. For example, H3K18ac and H3K4me3 levels were not sensitive to cell cycle modulation (Figs 1A–1D, 1I–1L, 1U–1W, and S4A), and neither were H4K8ac, H3K9ac, H3K18 crotonylation, total H3K9 methylation nor levels of total lysine acetylation (S3 and S4B Fig). Yet, strikingly, we found that H3K27ac levels were strongly reduced in cell cycle arrested genotypes, whereas levels of this modification were elevated when the cell cycle was accelerated (Fig 1E–1H and 1U–1W). In contrast, H3K9me3 and H3K27me3 levels were elevated in arrested cells, whereas levels of both modifications were reduced when the cell cycle was accelerated (Fig 1M–1W). Importantly, alterations observed in cell cycle arrested cells were independent of which cell cycle phase the arrest occurred in. Specifically, H3K27ac reduction and H3K9me3 and H3K27me3 elevation was observed in both G1 and G2 arrested cell populations upon expression of *pntP1* (Figs 1G, 1O, 1S; S1I and S1J), and we specifically confirmed downregulation of H3K27ac also in G1-arrested cells expressing *CycE-RNAi* or the p21 homologue *dacapo (dap)* (S2G–S2J Fig), suggesting that the elongation of either gap phase can produce the observed effects.

Taken together, we conclude that cell cycle dynamics have a surprising impact on histone modification in a proliferating tissue. First, only three tested modifications responded significantly to cell cycle dynamics in our assays, suggesting that only a subset of modifications is strongly linked to the cell cycle machinery. Second, these changes were not dependent to cells arresting in G1 or in G2, suggesting that overall cell cycle length is a defining factor rather than the gap phase position within a cell cycle. Third, if levels of histone modifications were primarily determined by the kinetics of restoring histone modifications after semi-conservative replication and the resulting histone PTM dilution, one may predict that modifications simply accumulate as the cell cycle lengthens. However, the reduction of H3K27ac in arrested cells defies a scenario in which continuous HAT activity restores H3K27ac levels after DNA replication. We thus wanted to better understand the relationship between histone methylation, histone acetylation and cell cycle dynamics.

**Cell cycle-dependent changes in H3K27ac, H3K27me3, or H3K9me3 levels occur predominantly at preexisting target loci**

To first identify the genomic regions affected by the cell cycle-dependent changes in bulk levels of H3K27ac, H3K27me3, and H3K9me3, we performed CUT&Tag analysis on wing imaginal discs expressing *Cdk1-RNAi* under the control of *rn-GAL4* for 24 hours. Although *Cdk1-RNAi* is expressed in only a subset of cells within the wing disc (see S1A Fig), limiting the magnitude of detectable differences and thus the sensitivity of this assay, we consistently observed a decrease in H3K27ac levels, and an increase in levels of both H3K27me3 and H3K9me3 modifications around peaks sites (Fig 2A–2D). These changes occurred at genomic regions already carrying these modifications in wild-type tissues, as evident

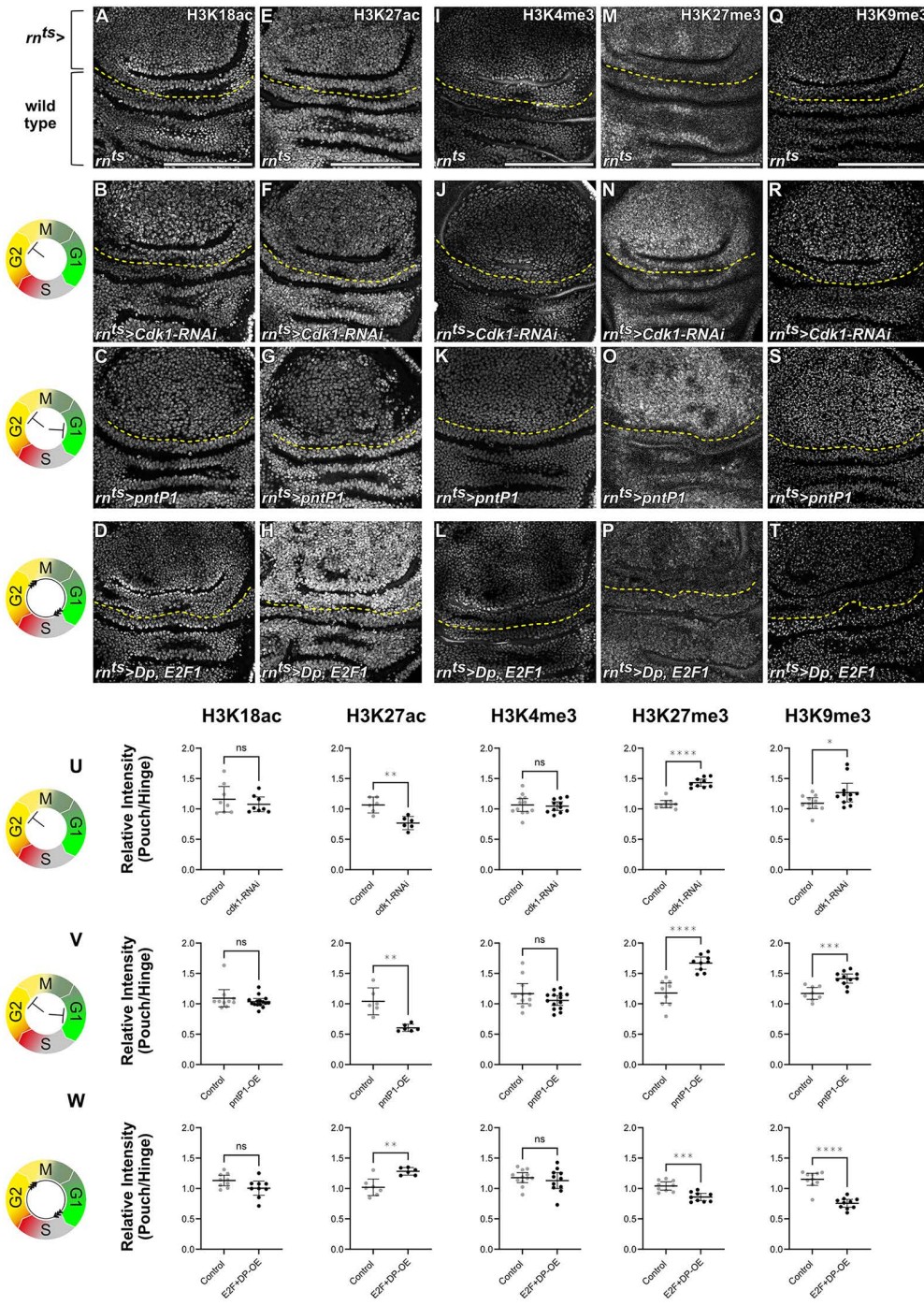

**Fig 1. H3K27ac, H3K27me3, and H3K9me3 levels respond to changes in cell cycle dynamics.** Icons represent the experimentally verified FUCCI cell cycle status in each condition: G2-phase arrest in *Cdk1-RNAi*-expressing discs, either G1 or G2-phase arrest in *pntP1*-expressing discs, and cell cycle acceleration in *Dp, E2F1*-overexpressing discs. For better guidance through subsequent figures, schemes already introduce FUCCI reporter read-outs for phase G1 (green), early S-phase (gray), late S-phase (red), and G2 (yellow). A–D. Immunostaining for H3K18ac in control (A), *Cdk1-RNAi*-expressing (B), *pntP1*-expressing (C), and *Dp, E2F1*-coexpressing (D) discs. Expression was induced for 24 h in the wing pouch using *rn-GAL4*. The *rn-GAL4* expressing pouch domain is represented by the tissue above the yellow line (rn^ts>). The tissue below the yellow line represents wild-type tissues of the hinge. E–T. Immunostaining for H3K27ac (E–H), H3K4me3 (I–L), H3K27me3 (M–P), H3K9me3 (Q–T) in control (E, I, M, Q), *Cdk1-RNAi*-expressing (F, J, N, R), *pntP1*-expressing (G, K, O, S) and *Dp, E2F1*-coexpressing (H, L, P, T) discs. **U.** Quantification of relative

signal intensities for H3K18ac, H3K27ac, H3K4me3, H3K27me3, and H3K9me3, presented as pouch-to-hinge ratios (i.e., rn$^{ts}$-to-wild-type-cell ratio) in *Cdk1-RNAi*-expressing discs. Mean and 95% CI is shown. Statistical significance was tested using two-tailed Mann–Whitney test for H3K18ac (control discs: $n = 8$, *Cdk1-RNAi*-expressing discs: $n = 8$); two-tailed Unpaired *t* test for H3K27ac (control discs: $n = 6$, *Cdk1-RNAi*-expressing discs: $n = 6$); two-tailed Unpaired *t* test for H3K4me3 (control discs: $n = 12$, *Cdk1-RNAi*-expressing discs: $n = 11$); two-tailed Mann–Whitney test for H3K27me3 (control discs: $n = 9$, *Cdk1-RNAi*-expressing discs: $n = 9$); two-tailed Mann–Whitney test for H3K9me3 (control discs: $n = 11$, *Cdk1-RNAi*-expressing discs: $n = 11$). **V.** Quantification of relative signal intensities for H3K18ac, H3K27ac, H3K4me3, H3K27me3, and H3K9me3, presented as pouch-to-hinge ratios (i.e., rn$^{ts}$-to-wild-type-cell ratio) in *pntP1*-expressing discs. Mean and 95% CI is shown. Statistical significance was tested using two-tailed Mann–Whitney test for H3K18ac (control discs: $n = 10$, *pntP1*-expressing discs: $n = 15$); two-tailed Welch's *t* test for H3K27ac (control discs: $n = 6$, *pntP1*-expressing discs: $n = 6$); two-tailed Unpaired *t* test for H3K4me3 (control discs: $n = 11$, *pntP1*-expressing discs: $n = 14$); two-tailed Unpaired *t* test for H3K27me3 (control discs: $n = 9$, *pntP1*-expressing discs: $n = 9$); two-tailed Unpaired *t* test for H3K9me3 (control discs: $n = 8$, *pntP1*-expressing discs: $n = 11$). **W.** Quantification of relative signal intensities for H3K18ac, H3K27ac, H3K4me3, H3K27me3, and H3K9me3, presented as pouch-to-hinge ratios (i.e., rn$^{ts}$-to-wild-type-cell ratio) in *Dp, E2F1*-coexpressing discs. Mean and 95% CI is shown. Statistical significance was tested using two-tailed Unpaired *t* test for H3K18ac (control discs: $n = 9$, *Dp, E2F1*-coexpressing discs: $n = 9$); two-tailed Unpaired *t* test for H3K27ac (control discs: $n = 7$, *Dp, E2F1*-coexpressing discs: $n = 6$); two-tailed Unpaired *t* test for H3K4me3 (control discs: $n = 12$, *Dp, E2F1*-coexpressing discs: $n = 11$); two-tailed Unpaired *t* test for H3K27me3 (control discs: $n = 9$, *Dp, E2F1*-coexpressing discs: $n = 9$); two-tailed Mann–Whitney test for H3K9me3 (control discs: $n = 10$, *Dp, E2F1*-coexpressing discs: $n = 10$). Fluorescence intensities are reported as arbitrary units. Sum projections of multiple confocal sections are shown in C, K, M, N, O, and Q–R. Maximum projections of multiple confocal sections are shown in A-B, E-F, G, H, I-J, P, S, and T. Scale bars: 100 μm. Please refer to (S4A Fig) for corresponding DAPI images of all discs shown. See S1 File for underlying data and statistical information.

from significant changes in average levels of H3K27ac, H3K27me3, or H3K9me3 modification in peaks called in both wild-type and *Cdk1-RNAi*-expressing discs (see 'peaks shared by WT and Cdk1-RNAi', Figs 2E–2G; S5A–S5C). These findings are consistent with prior observations in mESCs, where H3K27me3 levels increase at pre-existing sites during prolonged G1 phases [70]. Of note, many H3K27ac peaks identified only in wild-type samples exhibited a significant reduction in H3K27ac levels in *Cdk1-RNAi*-expressing discs, confirming that H3K27ac was indeed lost from preexisting acetylated regions (see 'peaks unique to WT' for H3K27ac, Figs 2E, 2F; and S5A, S5B, and S5D). Conversely, many H3K27me3 and H3K9me3 peaks were identified only in *Cdk1-RNAi*-expressing discs suggesting the potential occurrence or spreading to de novo methylation sites (see 'peaks unique to Cdk1-RNAi' for H3K27me and H3K9me3, Figs 2E, 2F; S5A, S5B, and S5E). We conclude that cell cycle-dependent changes to bulk levels of H3K27ac, H3K27me3, or H3K9me3 predominantly affect existing target sites; however, the increase specifically in H3K27me3 and H3K9me3 may also affect additional genomic regions. Importantly, the magnitude of H3K27me3 or H3K9me3 increase is similar for shared, unique, and nonpeak regions, suggesting that the increase in methylation occurs at the same rate across these domains (Fig 2E).

Our conclusions that a change in PTM levels mainly occur at preexisting loci is further supported by our observations that the histone-modifying enzymes responsible for H3K27 acetylation and H3K27 trimethylation do not compete for the same histone residues in vivo. Specifically, when we performed a knockdown of the H3K27 methyltransferase E(z), we observed an expected loss of H3K27me3 (S6A and S6B Fig), but importantly, no corresponding increase in H3K27ac (S6C and S6D Fig). Conversely, when we performed a knockdown of the H3K27 acetyltransferase CBP/Nej (Nejire), we observed an expected loss of H3K27ac (S6E and S6F Fig), but no corresponding increase in H3K27me3 (S6G and S6H Fig). We conclude that, even though studies propose local antagonism of H3K27me3 and H3K27ac at the same residues [34,37,71], the cell cycle-length-dependent decrease of H3K27ac and increase of H3K27me3 bulk levels upon expression of *Cdk1-RNAi* are not driven by competition for the same H3K27 residues but instead reflect the natural occurrence of these distinct PTMs in distinct genomic contexts.

### H3K27 or H3K9 modifications are linked to cell cycle dynamics during development

If bulk histone modifications in wing discs are affected by cell cycle progression, can we provide evidence that this also occurs in a physiologically relevant state? During larval development, cells in the wing imaginal disc actively proliferate. However, the future wing margin forms a zone of nonproliferating cells (ZNC), where cells arrest in either G1 or G2

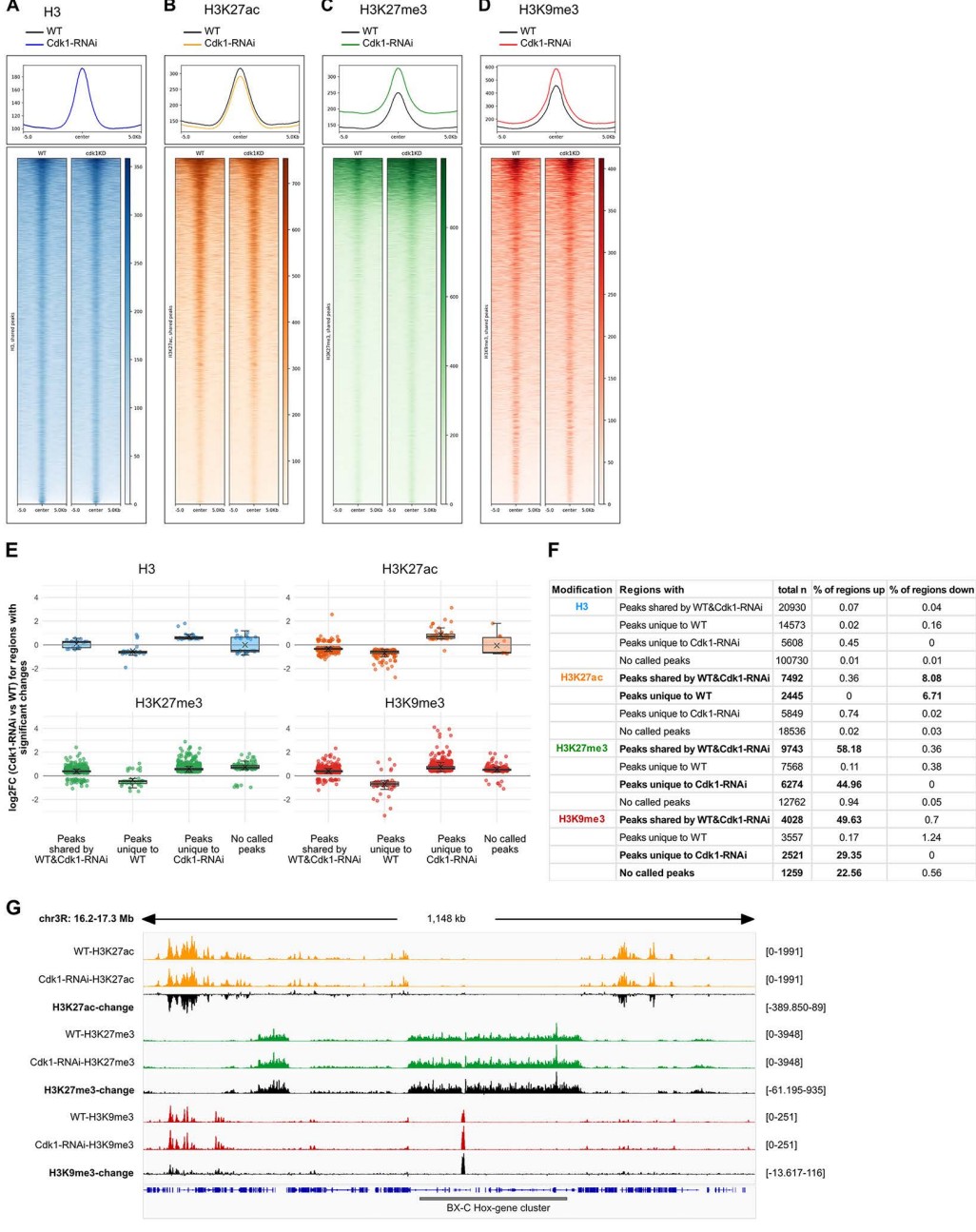

**F**

| Modification | Regions with | total n | % of regions up | % of regions down |
|---|---|---|---|---|
| H3 | Peaks shared by WT&Cdk1-RNAi | 20930 | 0.07 | 0.04 |
| | Peaks unique to WT | 14573 | 0.02 | 0.16 |
| | Peaks unique to Cdk1-RNAi | 5608 | 0.45 | 0 |
| | No called peaks | 100730 | 0.01 | 0.01 |
| H3K27ac | Peaks shared by WT&Cdk1-RNAi | 7492 | 0.36 | 8.08 |
| | Peaks unique to WT | 2445 | 0 | 6.71 |
| | Peaks unique to Cdk1-RNAi | 5849 | 0.74 | 0.02 |
| | No called peaks | 18536 | 0.02 | 0.03 |
| H3K27me3 | Peaks shared by WT&Cdk1-RNAi | 9743 | 58.18 | 0.36 |
| | Peaks unique to WT | 7568 | 0.11 | 0.38 |
| | Peaks unique to Cdk1-RNAi | 6274 | 44.96 | 0 |
| | No called peaks | 12762 | 0.94 | 0.05 |
| H3K9me3 | Peaks shared by WT&Cdk1-RNAi | 4028 | 49.63 | 0.7 |
| | Peaks unique to WT | 3557 | 0.17 | 1.24 |
| | Peaks unique to Cdk1-RNAi | 2521 | 29.35 | 0 |
| | No called peaks | 1259 | 22.56 | 0.56 |

**Fig 2. Cell cycle-dependent changes in H3K27ac, H3K27me3, and H3K9me3 occur mainly at preexisting genomic sites. A–D.** Profile plots and heatmaps showing normalized CUT&Tag signals at peaks shared by wild-type control and *Cdk1-RNAi*-expressing discs for H3 **(A)**, H3K27ac **(B)**, H3K27me3 **(C)**, and H3K9me3 **(D)** (see Materials and methods). Averaged and individual signal intensities are centered ±5 kb around the shared peak. **E.** Each box plot shows the distribution of log2 fold change values (Cdk1-RNAi relative to WT) for genomic regions grouped by peak category: peaks shared between WT and Cdk1-RNAi, peaks unique to WT, peaks unique to Cdk1-RNAi, and 500-bp regions with no called peaks in either condition. The x-axis indicates region category and the y-axis shows the log2 fold change of normalized counts (Cdk1-RNAi/WT). Points represent individual peaks or 500-bp genomic regions and are colored according to histone modifications (H3, blue; H3K27ac, orange; H3K27me3, green; H3K9me3, red). Crosses indicate the mean log2 fold change within each group. Only regions with basemean of more than 25 read counts and *DES*eq2 adjusted *p* < 0.1 are plotted. **F**. Table summarizes the total number of genomic regions described in (E) tested for each histone modification (H3, H3K27ac, H3K27me3, and H3K9me3) and the percentage of these regions that are significantly upregulated (log2FoldChange> 0) or significantly downregulated (log2Fold-Change < 0) in Cdk1-RNAi relative to WT. **G.** Example IGV genome browser snapshot of CUT&Tag profiles comparing H3K27ac, H3K27me3, and H3K9me3 levels in WT wing discs and discs expressing *Cdk1-RNAi* for 24 h in the pouch region under *rn-GAL4* control. Tracks for H3K27ac (orange),

H3K27me3 (green), and H3K9me3 (red) were normalized to H3 and spike-in signals as described in the Materials and Methods and visualized in IGV. Black difference tracks show the change between conditions, calculated as Cdk1-RNAi minus WT read counts for each histone modifications. Track also covers the BX-C Hox gene cluster encoding Ubx, Abd-A and Abd-B targeted by Polycomb silencing. Access to underlying data is provided at NCBI (SRA) PRJNA130380. See Code and Data Availability Statement.

(magenta bracket in Figs 3A, 3B and S7A) [72,73]. In addition to the ZNC, two clusters of G2-arrested cells—one with senescent features—can be found in the hinge domain (arrowheads in Figs 3A–3C and S7B) [74]. Strikingly, arrested cells in the ZNC (Figs 3D–3F, 3J and S7G) and the hinge (Figs 3G–3J and S7G) showed a decrease in H3K27ac levels and an increase in H3K9me3 and H3K27me3, consistent with our previous conclusions upon cell cycle elongation by expression of *Cdk1-RNAi* or *UAS-pntP1*. Importantly, other histone modifications only showed weak alterations, again reinforcing the specific association between H3K9me3, H3K27me3, and H3K27ac levels and the dynamics of cell cycle progression (Figs 3J and S7C–S7G).

To begin to understand if there are consequences for cell cycle-dependent PTM modulation during development, we analyzed—as an example of the three PTMs—H3K27me3-dependent regulation of cell fates. H3K27me3 mediates Polycomb-dependent silencing of genes typically required for cell fate specification [24,25,27]. In imaginal disc with either arrested (*Cdk1-RNAi*) or accelerated (*Dp, E2F1*) cell cycle dynamics, the expression of Polycomb targeted genes required for wing disc differentiation, such as Antp, Ubx/AbdA, En/inv, Nub, Wg, Ptc, and Cut was not altered (S8 Fig). These observations demonstrate that reduced or elevated H3K27me3 levels observed upon changing cell cycle dynamics still ensure robust Polycomb target gene outputs, suggesting that a H3K27me3 levels buffer gene silencing against a range of cell cycle dynamics.

Surprisingly, our subsequent experiments revealed that the cell cycle-dependent regulation of H3K9me3, H3K27me3, and H3K27ac arises through mechanistically distinct pathways. Specifically, methylation and acetylation marks exhibit fundamentally different dynamics. Given these divergent mechanisms, we present our findings in two consecutive sections: first, we detail how H3K9me3 and H3K27me3 levels are coupled to the cell cycle via growth signaling pathways (Fig 4), followed by an independent analysis of cell cycle-dependent regulation of H3K27ac during S-phase (Fig 5).

## H3K27me3 or H3K9me3 are coupled to cell cycle dynamics via metabolic signaling

Previous studies suggest that H3K27me3 and H3K9me3 modifications are reestablished after DNA replication through slow and continuous methyltransferase activity in subsequent gap phases [9,10,47]. Our findings are consistent with these observations: in all our imaginal disc models, levels of H3K9me3 and H3K27me3 increase in conditions with elongated cell cycles (thus infrequent S-phase events, reducing replication-associated dilution of these marks), and they decrease in conditions with shortened cell cycles (increasing the frequency of S-phase and thereby enhancing dilution). These results suggest that H3K9 and H3K27 trimethylation is driven by a relatively constant rate of histone methyltransferase activity, which appears not to be affected by genetic alterations of the cell cycle machinery. If the rate of histone methylation can indeed be genetically separated from the frequency of cell cycling, what mechanisms then govern the restoration of histone methylation to the appropriate levels, particularly for H3K9me3, which is essential for heterochromatin architecture, and H3K27me3, which is critical for gene silencing? To address this question, we decided to manipulate cell cycle progression upstream of direct regulators of the cell cycle by activating a proliferative pathway with important metabolic functions. Specifically, we expressed a constitutively active Insulin receptor (InR[DA]) under the control of *rn-GAL4* for 24 h, which promotes PI3K/Akt signaling. The resulting metabolic changes can be visualized by higher rates of protein synthesis reflecting higher biosynthetic capacity, accelerated cell proliferation visualized by high EdU incorporation and ultimately, larger wing pouch size (Figs 4A–4C and S9A–S9E) (see also S8 Fig in [63]). Of note, expression of InR[DA] did not visually alter the FUCCI cell cycle profile (S9F and S9G). Combined these results demonstrate that the higher metabolic activity accelerates the cell cycle and thus proliferation, without pronounced alterations to the phase distribution.

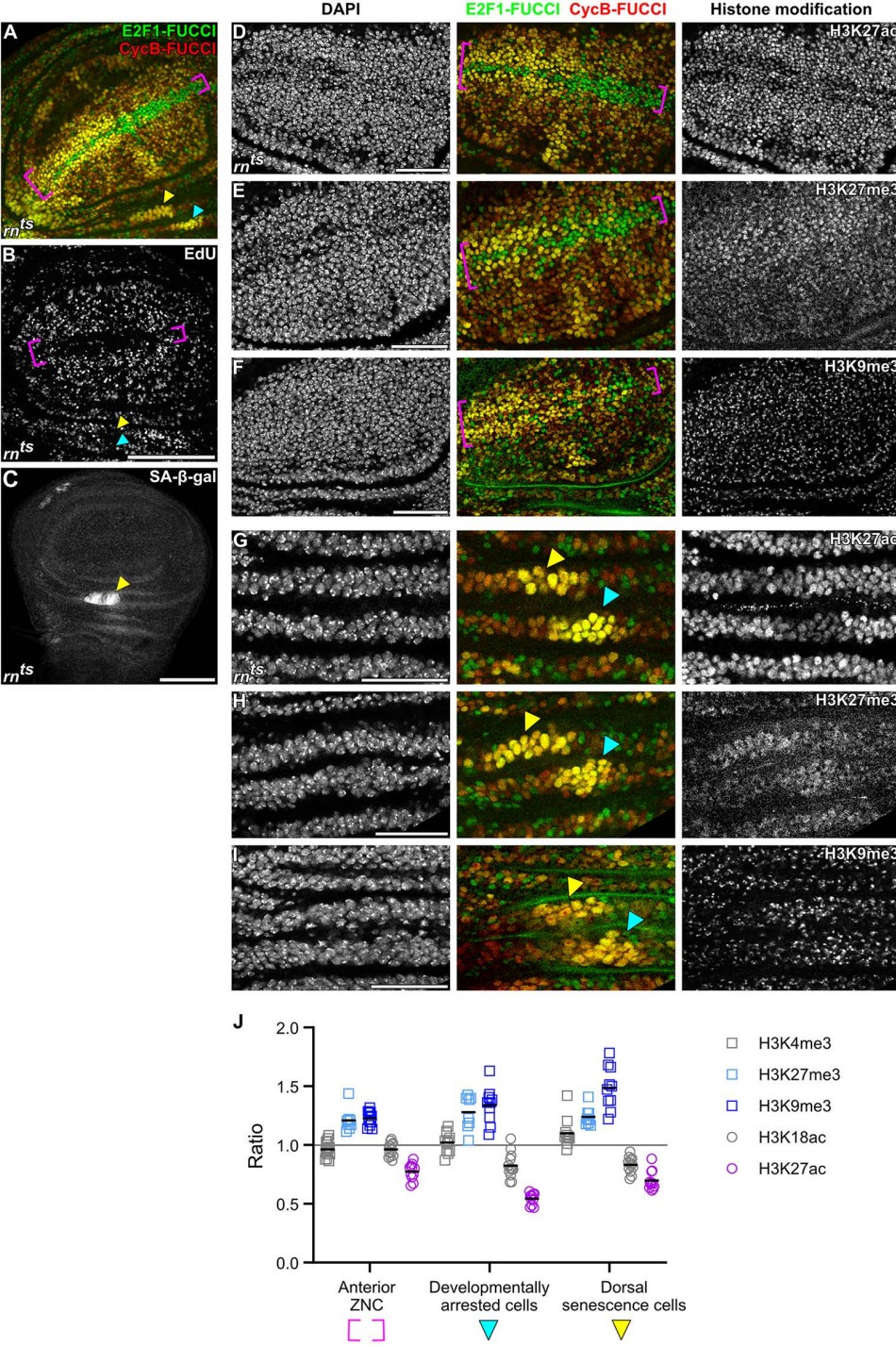

**Fig 3. Developmentally arrested cell populations under physiological conditions exhibit changes in H3K27ac, H3K27me3, and H3K9me3 levels.** A and B. The Fly-FUCCI system (A) and EdU incorporation assay (B) were used to visualize the spatial pattern of cell cycle arrest in zone of non-proliferating cells (ZNC) (magenta brackets), developmentally arrested cells (cyan arrowheads) and dorsal senescence cells (yellow arrowheads). In the anterior ZNC, central margin cells arrest in G1, while adjacent cells arrest in G2. Cells in the posterior ZNC uniformly arrest in G1. Developmentally arrested cells and dorsal senescence cells in the hinge region arrest in G2. Cell cycle arrested regions lack EdU incorporation, indicating absence of S-phase entry. **C.** Elevated SA-β-gal activity marks dorsal senescence cells undergoing programmed senescence (yellow arrowheads) in developing wing discs. D–F. Immunostaining for H3K27ac (D), H3K27me3 (E), and H3K9me3 (F) in the ZNC of developing wing discs. Cells in the ZNC (magenta

brackets) are arrested in either G1 or G2 phase, and can be identified by the Fly-FUCCI reporters *GFP-E2F1*[1-230] (green) and *mRFP-NLS-CycB*[1-266] (red). Among the histone modifications examined, only H3K27ac, H3K27me3, and H3K9me3 exhibit cell cycle arrest associated changes in the ZNC. G–I. Immunostaining for H3K27ac (G), H3K27me3 (H), and H3K9me3 (I) in the wing disc hinge region; dorsal senescence cells are indicated by yellow arrowheads and developmentally arrested cells are indicated by cyan arrowheads. **J.** Quantified ratios of histone modification levels within the anterior ZNC, developmentally arrested cells, and dorsal senescence cells relative to regions outside of these domains (see Materials and methods and S7G Fig for details) in normally developing wing discs. Each dot represents an individual wing imaginal disc; symbols/colors correspond to the indicated histone modifications, and horizontal bars indicate the mean ratio for each group. Discs were stained with DAPI to visualize nuclei. Scale bars: 100 μm in B and C; 50 μm in D–I. See S1 File for underlying data and statistical information.

Importantly, we found that expression of InR[DA] alone did not give rise to changes in H3K27 or H3K9 trimethylation levels, despite the accelerated cell divisions (Fig 4D, 4E, 4H, 4I, 4P, and 4Q). This demonstrates that metabolic potential can tune the rate of H3K27 and H3K9 trimethylation, to keep levels of H3K27 or H3K9 trimethylation comparable to wild-type, despite accelerating cell cycle progression. To now functionally separate metabolic potential from cell cycle progression, we also arrested the cell cycle by coexpressing the constitutively active Insulin receptor with a *Cdk1-RNAi*. As expected from our previous experiments, levels of H3K27me3 and H3K9me3 increased, but most strikingly, levels of H3K27me3 and H3K9me3 were even higher than in discs expressing *Cdk-1-RNAi* alone (Fig 4F, 4G, 4J, 4K, 4P, and 4Q). These results demonstrate that deposition rate of H3K27 or H3K9 trimethylation is tightly coupled to cellular metabolic capacity, enabling trimethylation to keep pace with cell cycle progression and thereby counteract replication-associated dilution during S-phase (S9H and S9I). Of note, levels of H3K4me3 were not affected by these manipulations (Fig 4L–4O and 4R), indicating that a modification maintaining steady levels of bulk transcriptional activity is dynamic and may not be linked to metabolic rates or the rate of cell cycle progression [75].

We wanted to better understand the functional implications of our findings. The link between levels of H3K9me3 and cell cycle progression becomes relevant in physiological and pathological setting of senescence, where cytotoxic stress or oncogenic mutations provide metabolic activation that is juxtaposed to a senescent cell cycle arrest program. Accordingly, in senescent G2-arrested cells at the center of inflammatory tissue damage in wing imaginal discs induced by ectopic expression of the TNFα homologue eiger (egr) (S10A–S10F Fig) [64,76–78], we observed high levels of H3K9 trimethylation (S10G–S10I Fig, purple). Importantly, levels in senescent cells with inflammatory metabolism were higher than in quiescent cells at the disc periphery, which exhibit low insulin signaling activity (S10G–S10I Fig, yellow) [79]. These observations support our conclusions that high and low metabolic activity coupled to cell cycle arrest produce different levels of H3K9me3. Finally, in agreement with our previous conclusions, we found that cells undergoing fast regenerative proliferation around the site of inflammatory damage have lowest levels of H3K9me3 (S10G–S10I Fig, cyan).

## Opposing Nejire and HDAC1 activities shape H3K27ac dynamics across S-phase

Having revealed how H3K9me3 and H3K27me3 dynamics are coupled via metabolic cues to the cell cycle, we now wanted to characterize the observed cell cycle dynamics of H3K27 acetylation. The downregulation of H3K27ac by cell cycle elongation and the upregulation of H3K27ac by cell cycle acceleration contradicts the simple explanation that an enzymatic activity acetylates H3K27 at a constant rate. To better understand H3K27ac dynamics, we performed a detailed analysis of H3K27ac levels throughout the cell cycle by correlating H3K27ac with FUCCI reporters in wing imaginal discs. We specifically excluded mitotic cells from this analysis, as the highly condensed nature of their chromosomes and the cells' extreme apical position prevented a direct comparison to other cell cycle stages. Together, this approach allowed us to resolve H3K27ac dynamics across G1, early S, late S and G2 in vivo.

Using this approach, we found that H3K27ac levels were significantly increased in nuclei of early and late S-phase cells, if compared to G1 and G2 phases (Fig 5A–5C). More specifically, H3K27ac initially followed bulk acetylation and bulk H3 dynamics, which were highest in early S-phase (Figs 5D and S11B–S11F). However, while bulk acetylation and H3 levels sharply declined towards late S-phase, we observed that H3K27ac remained elevated before finally declining

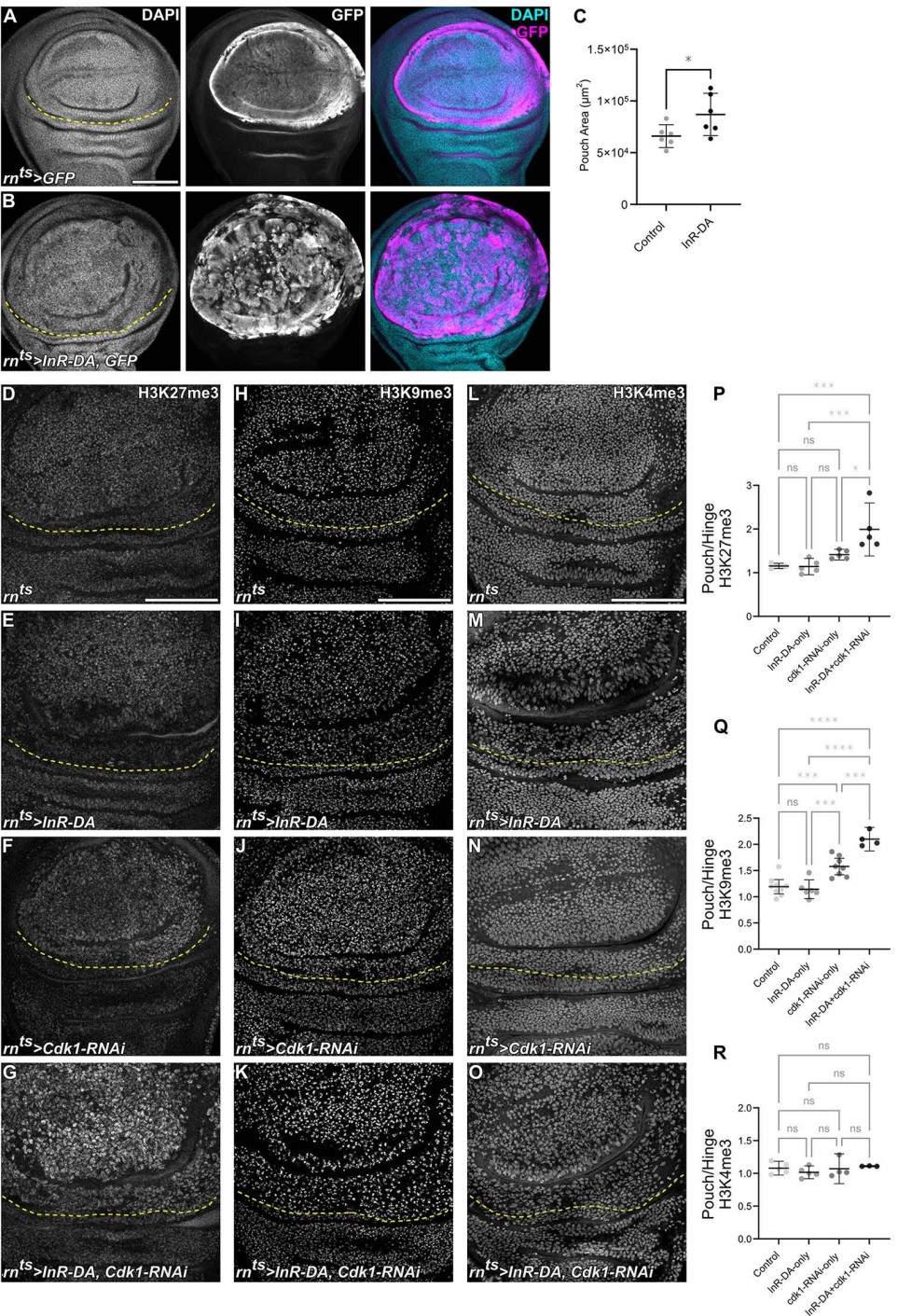

**Fig 4. Metabolic state sets the rate of H3K27 and H3K9 trimethylation.** A and **B.** Control wing disc expressing GFP (magenta) for 24 h in the wing pouch domain under the control of *rn-GAL4* (A). Wing disc expressing GFP and a constitutively active Insulin receptor (*InR-DA*) under the same conditions (B). GFP-expression visualizes the enlarged wing pouch domain in *InR-DA* expressing discs. **C.** Quantification of pouch size area in control and *InR-DA*-expressing discs. Mean and 95% CI is shown. Statistical significance was tested using two-tailed Unpaired *t* test. (control discs: *n* = 6, *InR-DA*-expressing discs: *n* = 6). D–G. Immunostaining for H3K27me3 in control (D), *InR-DA*-expressing (E), *Cdk1-RNAi*-expressing (F) and *InR-DA, Cdk1-RNAi*-coexpressing (G) discs. **H–K.** Immunostaining for H3K9me3 in control **(H)**, *InR-DA*-expressing **(I)**, *Cdk1-RNAi*-expressing **(J)** and *InR-DA, Cdk1-RNAi*-coexpressing **(K)** discs. L–O. Immunostaining for H3K4me3 in control (L), *InR-DA*-expressing (M), *Cdk1-RNAi*-expressing (N) and *InR-DA, Cdk1-RNAi*-coexpressing (O) discs. Expression was induced for 24 h in the wing pouch using *rn-GAL4.* Yellow dashed lines indicate the

boundary between the wing pouch and hinge regions. The area above the line corresponds to the *rn-GAL4* expression domain, while the area below represents wild-type cells. P–R. Quantification of relative signal intensities for H3K27me3 (P), H3K9me3 (Q) and H3K4me3 (R) presented as pouch-to-hinge ratios (i.e., rn$^{ts}$-to-wild-type-cell ratio) in *InR-DA*-expressing, *Cdk1-RNAi*-expressing and *InR-DA, Cdk1-RNAi*-coexpressing discs. Mean and 95% CI is shown. Statistical significance was tested using Ordinary one-way ANOVA followed by Tukey's post-hoc test for multiple comparison. For H3K27me3 control discs: $n = 5$, *InR-DA*-expressing discs: $n = 5$, *Cdk1-RNAi*-expressing discs: $n = 5$ and *InR-DA, Cdk1-RNAi*-coexpressing discs: $n = 5$. For H3K9me3 control discs: $n = 9$, *InR-DA*-expressing discs: $n = 6$, *Cdk1-RNAi*-expressing discs: $n = 8$ and *InR-DA, Cdk1-RNAi*-coexpressing discs: $n = 4$. For H3K4me3 control discs: $n = 5$, *InR-DA*-expressing discs: $n = 5$, *Cdk1-RNAi*-expressing discs: $n = 4$ and *InR-DA, Cdk1-RNAi*-coexpressing discs: $n = 3$. DAPI normalization was applied to H3K9me3 dataset to account for apparent signal dilution caused by increased tissue folding in the InR-DA+cdk1-RNAi genotype. Discs were stained with DAPI to visualize nuclei. Fluorescence intensities are reported as arbitrary units. Sum projections of multiple confocal sections are shown in A and B. Maximum projections of multiple confocal sections are shown in D–K. Scale bars: 100 µm. See S1 File for underlying data and statistical information.

in G2 (Figs 5C, 5D, and S11D). Of note, fluctuation of H3 and total lysine acetylation in our microscopy assays can be mostly explained by S-phase dependent histone synthesis and histone acetylation, as well as fluctuating morphology of interphase nuclei [59,80,81]. Our finding that H3K27ac levels are higher throughout early and late S-phase is consistent with our earlier observations: cells which cycle fast repeatedly enter S-phase and cells which cycle slowly are not often found in S-phase, hence bulk H3K27ac levels in a proliferating and arrested tissue will appear high and low, respectively (Fig 1H). Importantly, many fast-cycling *dE2F1,dDP*-expressing cells are in late S-phase, enhancing the specific effect of high H3K27ac during late S-phase in this genotype (S2E Fig). Combined, these observations identify H3K27ac as a novel S-phase regulated histone mark and whose dynamics deviate from global histone acetylation, consistent with a residue-specific cell cycle program.

To begin to understand how H3K27 acetylation may be regulated throughout the cell cycle, we first knocked down the H3K27ac HAT CBP/nej by expressing a *neijire-RNAi* for 24 h under the control *rn-GAL4*. This completely abolished H3K27ac staining in the wing pouch, confirming that CBP/nej is the main source of H3K27 acetylation in the wing disc (Fig 5E–5F) [32]. Strikingly, all *CBP/nej-RNAi*-expressing cells completely arrest in G2 within 24 h and do not reenter S-phase (Figs 5G–5H and S12P–S12R). From these observations, we conclude that H3K27 acetylation completely turns over within one cell cycle and that H3K27 acetylation within one cell cycle is completely dependent on CBP/nej. Importantly, levels of other CBP/nej-mediated modification (H3K18ac, H4K8ac, and H3K18 crotonylation) were not as sensitive to the loss of CBP/nej within one cell cycle, as these modifications in arrested *CBP/nej-RNAi*-expressing cells persisted, albeit at reduced levels (S12G–S12N Fig). Moreover, even though H3K18ac and H4K8ac showed a detectable increase in early S-phase in proliferating wild-type tissue, their levels did not remain high in late S-phase (S12A–S12F Fig). Combined, these observations indicate that H3K27ac displays unusually rapid, cell cycle-linked turnover and is fully dependent on CBP/nej within a single cycle, in contrast to other CBP/nej-mediated marks.

The pronounced increase of H3K27ac in early S-phase and its reduction towards late S-phase and G2 imply that both addition and removal of H3K27ac are tightly controlled during S-phase. Of note, a CBP/nej-GFP fusion protein expressed from the endogenous locus [82] displayed uniform nuclear GFP levels. This suggests that S-phase-linked H3K27ac dynamics are unlikely to be explained by cell cycle-dependent changes in CBP/nej protein abundance (S12O Fig). Indeed, in cultured cells, CBP/p300 activity may be regulated by CDK at the G1/S transition [83]. Together, these observations argue that cell cycle-dependent regulation of CBP/nej activity, rather than CBP/nej levels, contributes to the S-phase enrichment of H3K27ac. To better support this conclusion, we designed experiments to manipulate H3K27 acetylation and deacetylation dynamics and to correlate H3K27ac levels with cell cycle phases to reveal which features of H3K27ac dynamics reflect regulated kinetics rather than enzyme abundance.

We first overexpressed CBP/nej for 24 h under the control of *rn-GAL4*. This causes a pronounced increase in absolute levels H3K27ac in all cells (Figs 5I, 5J, 5O; S11G and S11H). Strikingly, despite higher levels of CBP/nej and H3K27ac, the cell cycle-dependent dynamics of H3K27ac were maintained, suggesting strong regulatory inputs that preserve

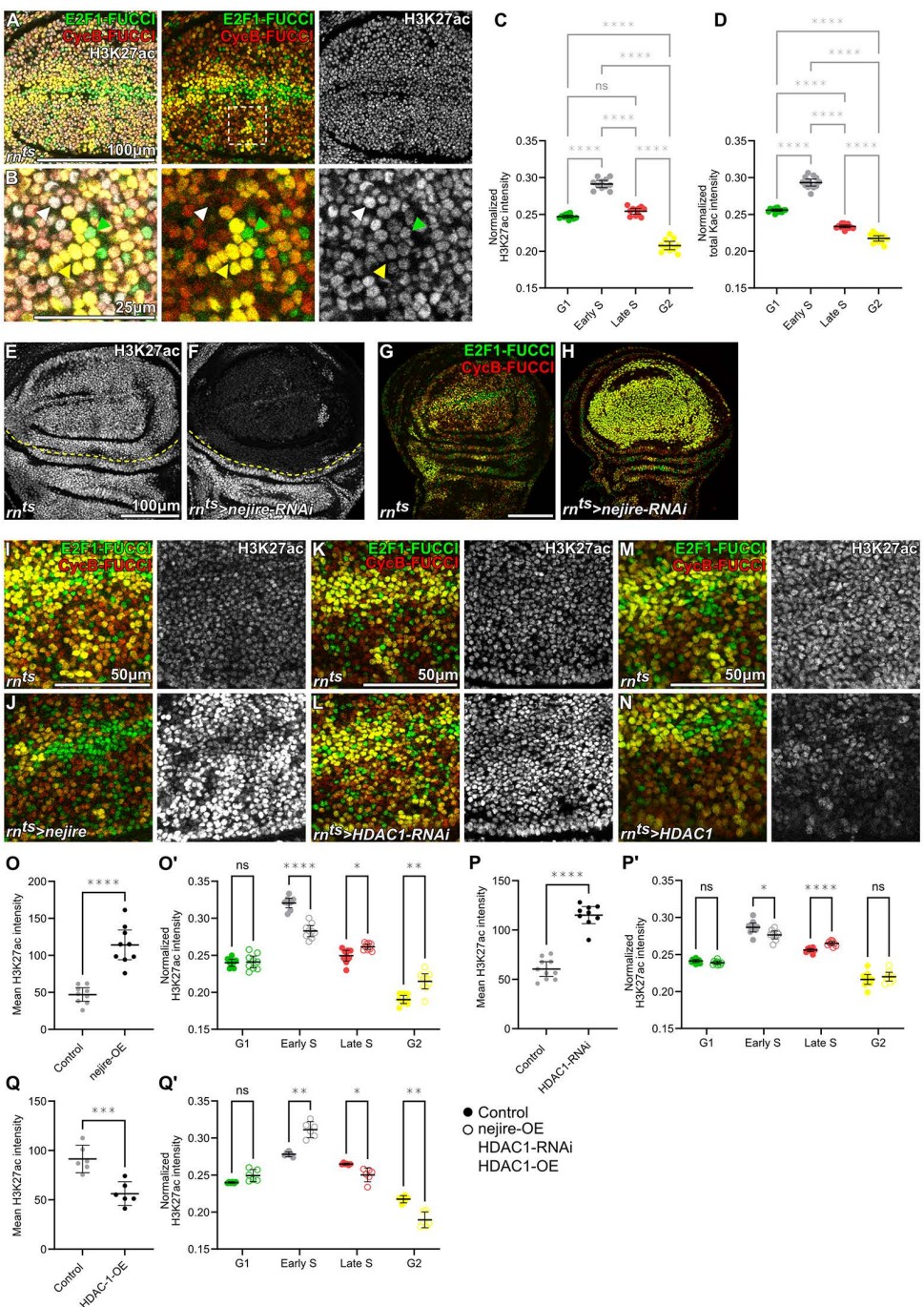

**Fig 5. Opposing Nejire and HDAC1 activities shape H3K27ac dynamics across S-phase.** A and **B.** Immunostaining for H3K27ac in a control disc expressing the FUCCI reporters *GFP-E2F1*[1-230] (green) and *mRFP-NLS-CycB*[1-266] (red) to visualize cell cycle phases (A). Magnified region in (B) is demarcated by a white dashed square in (A). **C.** Quantification of normalized H3K27ac intensity level in different cell cycle phases in the pouch region of control discs. Normalization was performed against the average fluorescence intensity across all cell cycle phases. Mean and 95% CI is shown. Statistical significance was tested using Repeated Measures One-Way ANOVA followed by Tukey's post-hoc test for multiple comparison (*n* = 11 discs). **D.** Quantification of normalized total acetylated lysine level in different cell cycle phases in the pouch region of control discs. Normalization was performed against the average fluorescence intensity across all cell cycle phases. Mean and 95% CI is shown. Statistical significance was tested using Repeated Measures One-Way ANOVA followed by Tukey's post-hoc test for multiple comparison (*n* = 13 discs). E and **F.** Immunostaining for H3K27ac in control (E) and *nejire-RNAi*-expressing (F) discs. Expression was induced for 24 h in the wing pouch using *rn-GAL4.* Yellow dashed lines indicate the boundary between the wing pouch and hinge regions. G–H. Cell cycle dynamics in control (G) and *nejire-RNAi*-expressing (H) discs. FUCCI

reporters *GFP-E2F1*[1-230] (green) and *mRFP-NLS-CycB*[1-266] (red) were used to visualize cell cycle phases. There is a shift of cells to the G2 phase of the cell cycle in *nejire-RNAi*-expressing wing pouch. I–J. Immunostaining for H3K27ac in control (I) and *nejire*-expressing (J) discs with FUCCI reporters *GFP-E2F1*[1-230] (green) and *mRFP-NLS-CycB*[1-266] (red) to visualize cell cycle phases. Expression of *nejire* was induced for 24 h in the wing pouch using *rn-GAL4*. K–L. Immunostaining for H3K27ac in control (K) and *HDAC1-RNAi*-expressing (L) discs with FUCCI reporters *GFP-E2F1*[1-230] (green) and *mRFP-NLS-CycB*[1-266] (red) to visualize cell cycle phases. Expression of *HDAC1-RNAi* was induced for 24 h in the wing pouch using *rn-GAL4*. M–N. Immunostaining for H3K27ac in control (M) and *HDAC1*-expressing (N) discs with FUCCI reporters *GFP-E2F1*[1-230] (green) and *mRFP-NLS-CycB*[1-266] (red) to visualize cell cycle phases. Expression of *HDAC1* was induced for 24 h in the wing pouch using *rn-GAL4*. **O.** Quantification of mean H3K27ac intensity in the pouch region of control or *nejire*-expressing discs. Mean and 95% CI is shown. Statistical significance was tested using two-tailed Welch's *t* test (control discs: $n = 9$, *nejire*-expressing discs: $n = 9$). O'. Quantification of normalized H3K27ac intensity level in different cell cycle phases in the pouch region of control or *nejire*-expressing discs. Normalization was performed to the average fluorescence intensity across all FUCCI-defined phases (to avoid privileging any single phase as the normalization reference). Mean and 95% CI is shown. Statistical significance was tested using Repeated Measures Two-Way ANOVA followed by with Šidák's post-hoc test for multiple comparisons (control discs: $n = 9$, *nejire*-expressing discs: $n = 9$). **P.** Quantification of mean H3K27ac intensity in the pouch region of control or *HDAC1-RNAi*-expressing discs. Mean and 95% CI is shown. Statistical significance was tested using two-tailed Mann–Whitney test. (control discs: $n = 10$, *HDAC1-RNAi*-expressing discs: $n = 9$). **P'.** Quantification of normalized H3K27ac intensity level in different cell cycle phases in the pouch region of control or *HDAC1-RNAi*-expressing discs. Normalization was performed to the average fluorescence intensity across all FUCCI-defined phases (to avoid privileging any single phase as the normalization reference). Mean and 95% CI is shown. Statistical significance was tested using Repeated Measures Two–Way ANOVA followed by with Šidák's post-hoc test for multiple comparisons (control discs: $n = 10$, *HDAC1*-expressing discs: $n = 9$). Q. Quantification of mean H3K27ac intensity in the pouch region of control or *HDAC1*-expressing discs. Mean and 95% CI is shown. Statistical significance was tested using two-tailed Unpaired *t* test (control discs: $n = 6$, *HDAC1*-expressing discs: $n = 6$). Q'. Quantification of normalized H3K27ac intensity level in different cell cycle phases in the pouch region of control or *HDAC1*-expressing discs. Normalization was performed to the average fluorescence intensity across all FUCCI-defined phases (to avoid privileging any single phase as the normalization reference). Mean and 95% CI is shown. Statistical significance was tested using Repeated Measures Two-Way ANOVA followed by with Šidák's post-hoc test for multiple comparisons (control discs: $n = 6$, *HDAC1*-expressing discs: $n = 6$). Fluorescence intensities are reported as arbitrary units. Sum projections of multiple confocal sections are shown in E and F. Scale bars: 100 μm in A, E and F and G and H; 50 μm in I and J, K and L, and M and N; 25 μm in B. See S1 File for underlying data and statistical information.

H3K27ac pattern across the cycle. However, the elevated baseline of H3K27ac resulted in a less pronounced rise in early S-phase and a less pronounced decline in late S-phase compared to wild-type dynamics (Fig 5O'). Importantly, this behavior was phenocopied by depletion of the primary H3K27 deacetylase HDAC1 (Rpd3) via expression of an RNAi construct under the control of *rn-GAL4* for 24 h [32]. HDAC1 knockdown strongly increased H3K27ac while maintaining its cell cycle-linked dynamics, and again elevated baseline levels dampened both the early S-phase increase and late S-phase decrease (Figs 5K, 5L, 5P, 5P'; S11I and S11J). These experiments demonstrate that cell cycle linked acetylation of H3K27 is robust to large shifts in basal acetylation state, but also demonstrates that H3K27ac accumulates ectopically in late S-phase cells when CBP/nej levels are too high or HDAC1 levels are too low.

These results are supported by the reverse experiment. We overexpressed HDAC1 to drive deacetylation, which caused a pronounced decrease in absolute H3K27ac levels in all cells (Figs 5M, 5N, 5Q; S11K and S11L). Strikingly, despite the much lower baseline, the cell cycle-dependent dynamics of H3K27ac were maintained. Moreover, the low baseline was associated with a more pronounced rise in early S-phase and a more pronounced decline in late S-phase compared to wild-type (Fig 5Q'). Combined, the disproportionate reduction of H3K27ac between early and late S-phase supports a model in which HDAC1-mediated removal is particularly effective at this stage. This suggests that deacetylation during late S-phase is a key driver of the late S-phase decline in H3K27ac levels.

## Discussion

Our work dissects how cell cycle dynamics shape chromatin states in a proliferating tissue in vivo. By accelerating or stalling the cell cycle in the wing pouch, we found that bulk levels of only three prominent histone modifications consistently tracked with cell cycle length: H3K27ac decreased upon cell cycle stalling and increased upon acceleration, whereas H3K27me3 and H3K9me3 behaved in the opposite direction. These effects were independent of whether cells were arrested in G1 or G2, indicating that overall cell cycle length, rather than the identity of a particular phase, is a defining parameter. Thus, our data reveal an unexpectedly selective link between cell cycle dynamics and a small but prominent set of histone modifications.

Genome-wide analyses using CUT&Tag assays revealed that these changes in H3K27ac, H3K27me3, and H3K9me3 occur predominantly at their distinct pre-existing genomic target loci, implicating canonical histone readers and modifiers in mediating these cell cycle-dependent changes. Importantly, despite cell cycle-dependent changes in bulk H3K27me3 levels, expression patterns of Polycomb target genes remained unaffected, implying that H3K27me3-dependent gene silencing at target loci is buffered even under altered cell cycle conditions. These findings are consistent with prior studies reporting largely normal cell fate determination in cell cycle mutants [60,84–86]. Importantly, fate changes have only been reported in the context of compensatory proliferation after tissue damage, where the strong concurrent activation of damage-response pathways such as JNK, JAK/STAT, or Wnt facilitates transdetermination [87–93]. Thus, cell cycle-dependent shifts in Polycomb-regulated modifications appear to be buffered under homeostatic conditions, whereas damage-associated signaling can convert altered proliferative states into changes in cell identity.

Our data are consistent with a framework in which H3K27me3 and H3K9me3 are restored gradually after replication and their steady-state levels reflect a balance between replication-coupled dilution and ongoing methyltransferase activity. In this view, elongating the cell cycle reduces the frequency of S-phase dilution events and permits accumulation, whereas accelerating the cycle increases dilution frequency and lowers steady-state levels. This in vivo behavior is consistent with findings in diverse cultured cell types that H3K27me3 accumulates during experimentally lengthened G1-phases or upon G1/G0 arrest [9,70], supporting a general principle that the timing between replication events can tune steady-state H3K27me3. We found that activating insulin signaling uncoupled cell cycle acceleration from methylation loss: insulin receptor activation increased proliferation without reducing H3K27me3 or H3K9me3 levels, implying that enhanced metabolic capacity can increase methylation rates to match the increased frequency of replication. Conversely, when elevated insulin signaling was combined with a cell cycle arrest, levels rose far above those in arrested wild-type tissue, revealing that effective methylation rate can be strongly boosted by metabolic state when dilution is removed. Thus, our results support a model in which metabolic potential tunes the rate of H3K27 and H3K9 trimethylation to match replication-linked dilution, thereby stabilizing repressive chromatin across diverse proliferative regimes.

An important implication of our findings relates to the emergence of senescence-associated heterochromatin, a phenomenon well-documented during oncogene-induced senescence [5, 94,95]. Our data suggest that H3K9me3 accumulation in senescence-associated heterochromatin may reflect persistent methyltransferase activity driven by aberrant metabolic stimuli in the absence of replication-dependent dilution. Thus, the observed chromatin state in senescent cells may be a consequence of the mismatch between metabolic and cycling rate, rather than an intrinsic feature of the senescence phenotype itself. An important implication of our findings is that post-mitotic and differentiating cells must tightly regulate histone-modifying activities to prevent the aberrant accumulation of repressive marks, such as H3K9me3 and H3K27me3.

In contrast to methylation marks, H3K27ac behaved as a rapidly turning-over modification with pronounced cell cycle structure. Quantitative analysis revealed that H3K27ac is enriched in early and late S-phase relative to G1 and G2, and unlike bulk acetylation, it remains elevated into late S-phase before declining in G2. While hyperacetylation of newly synthesized histones at various residues during S-phase is well-established, an S-phase dependent increase in nuclear acetylation of H3K27 has not been reported [49–58]. We found that CBP/nej depletion eliminated H3K27ac within 24 h and caused a rapid G2 arrest, indicating that H3K27ac is fully CBP/nej-dependent and turns over within a single cell cycle. Perturbing the writer or eraser shifted baseline levels but preserved the temporal pattern: CBP/nej overexpression or HDAC1 depletion increased H3K27ac yet maintained S-phase-linked dynamics, while HDAC1 overexpression lowered baseline levels and accentuated the early-to-late S-phase drop. Together with uniform nuclear CBP/nej-GFP levels, these data argue that H3K27ac dynamics are not set simply by CBP/nej abundance but by cell cycle-regulated acetylation, with late S-phase deacetylation emerging as a dominant resetting step. Thus, H3K27ac is governed by a S-phase regulated acetylation and deacetylation, in which CBP/nej provides acetylation activity during early S-phase and HDAC1-dependent removal resets H3K27ac in late S-phase.

While CBP/nej has been reported to be regulated by CDK at the G1/S transition [83], HDAC1 is best known to undergo mitosis-associated phosphorylation regulating chromatin association and enzymatic activity during mitotic entry [96–99]. Several studies reported H3K27ac to be present at Origin Recognition Complex-binding sites and to be required for gene amplification during endoreplication [39,100–102], raising the possibility that elevated H3K27ac promotes an open chromatin environment that supports efficient DNA replication. In contrast, HDAC activity is implicated in removing hyper-acetylation from newly synthesized histones at the end of S-phase to reestablish higher order chromatin organization [103], suggesting that H3K27ac could be one residue targeted within this process. Together, these studies provide a plausible framework for our observation that H3K27ac is established during early S-phase and reset in late S-phase by deacetylation.

Collectively, our findings define two mechanistically separable ways in which cell cycle dynamics shape chromatin in vivo. Repressive methylation marks (H3K27me3 and H3K9me3) behave as slow-turnover modifications whose levels are set by the balance between replication-associated dilution and metabolically tuned methylation rates, providing a mechanism to maintain stable repression across changing proliferative states. In parallel, H3K27ac is controlled by a rapid S-phase-linked CBP/nej-HDAC1 cycle that is robust to large baseline shifts but sensitive to the timing of acetylation and deacetylation within S-phase. Together, these modules reveal how growth state and cell cycle progression are integrated to stabilize chromatin landscapes during development while also creating vulnerabilities in pathological states where metabolism and proliferation become uncoupled.

## Experimental procedures

### *Drosophila* genetics

All experiments were performed on *Drosophila melanogaster*. Fly strains (see S1 Table) were maintained on standard fly food (10 L water, 74,5 g agar, 243 g dry yeast, 580 g corn flour, 552 ml molasses, 20.7 g Nipagin, 35 ml propionic acid) at 18–22 °C. Larvae from experimental crosses were allowed to be fed on Bloomington formulation (175.7 g Nutry-Fly,1,100 ml water 20 g dry yeast, 1.45 g Nipagin in 15 ml Ethanol, 4.8 ml Propionic acid) and raised at 18 °C or 30 °C to control GAL80ts-dependent induction of GAL4/UAS. To drive expression of UAS-constructs, such as *UAS-Cdk1-RNAi*, under the control of *rn-GAL4* in the wing pouch, experiments were conducted following protocols from Smith-Bolton and colleagues [78], and Cosolo and colleagues [64], with slight modifications. Briefly, larvae with the genotype *rn-GAL4, tub-GAL80ts* and appropriate UAS-transgenes were staged by a 6 h egg collection and raised at 18 °C with a density of 50 larvae per vial. Transgene expression was activated by shifting the temperature to 30 °C for 24 h on day 7 after egg deposition (AED). Unless otherwise noted, dissections were performed immediately after this induction period (recovery time point R0). Control genotypes were derived from either rn^ts control crosses or sibling animals (+/TM6c). Our experimental design did not consider differences between sexes unless for genetic crossing schemes.

### Immunohistochemistry of wing imaginal discs

To visualize histone modifications, wing discs from third instar larvae were dissected and fixed for 10 min at room temperature in 4% paraformaldehyde in PBS. Washes were performed in PBS containing 0.1% TritonX-100 (PBT). The discs were then incubated with primary antibodies (S1 Table) in 0.1% PBT, gently mixing 2-overnights at 4 °C. During incubation with cross-absorbed secondary antibodies coupled to Alexa Fluorophores (S1 Table) at 4 °C for 1-overnight, tissues were counterstained with DAPI (0.25 ng/µL, Sigma, D9542). Tissues were mounted using SlowFade Gold Antifade (Invitrogen, S36936). To visualize DNA damage using the anti-H2AvD phosphoS137 antibody, wing discs were fixed 5 min at room temperature in 4% paraformaldehyde in PBS. Washes were performed in PBS containing 0.5% TritonX-100. The primary antibody was incubated in 0.5% PBT overnight at 4 °C and the secondary antibody in 0.5% PBT for 2 h at room temperature. For all other staining protocols, discs were fixed for 15 min at room temperature in 4% paraformaldehyde in PBS. All washes and incubations were performed in PBS containing 0.1% TritonX-100 as described above. To maintain

consistency in staining across genotypes, control and experimental wing discs were processed together in the same vial throughout the protocol and mounted on the same slide whenever possible. Images were acquired using the Leica TCS SP8 Microscope, using the same confocal settings for linked samples and processed using tools in Fiji. Figure panels were assembled in Affinity Designer 2.

### EdU labeling

EdU incorporation was performed using the Click-iT Plus EdU Alexa Fluor 647 Imaging Kit (S1 Table). Briefly, larval cuticles were inverted in Schneider's medium and incubated with EdU (10 µM final concentration) at RT for 15 min. Cuticles were then fixed in 4% PFA/PBS for 15 min, washed for 30 min in PBT 0.5%. EdU-Click-iT labeling was performed according to manufacturer's guidelines. Tissues were washed in PBT 0.1%, after which immunohistochemistry, sample processing, and imaging were carried out as described above.

### SA-β-Gal staining

CellEvent senescence detection kit from Invitrogen (S1 Table) was used to check senescence-associated β-galactosidase activity, following the manufacturer instructions. Briefly, larvae were dissected in PBS, fixed with 4% PFA, washed with 1% BSA (in PBS), and then incubated in working solution for 2 h at 37 °C. Washing steps were performed in PBS and PBS containing 0.1% TritonX-100 (PBT). Tissues were counterstained with DAPI (0.25 ng/µl). Further immunohistochemistry analysis and sample mounting were performed as described above.

### Protein synthesis assay using OPP- Click-iT staining

OPP Assays were performed using Click-iT Plus OPP Protein Synthesis Assay Kits (Invitrogen Molecular Probe) according to manufacturer's instructions. Briefly, larvae were dissected and inverted cuticles were incubated with a 1:1000 dilution of Component A in Schneider's medium for 15 min on a nutator. Larval cuticles were fixed with 4% paraformaldehyde for 15 min, rinsed twice in 0.1% PBT, and permeabilized with 0.5% PBT for 15 min. The cuticles were then stained with the Click-iT cocktail for 30 min at room temperature, protected from light. Further immunohistochemistry analysis and sample mounting were performed as described above.

### CUT&Tag sample preparation

Briefly, for each genotype around 80−100 wing discs were dissected in ice cold Schneider's medium, collected, snap frozen and stored at −80 °C until further use. Nuclei were isolated with the help of a mechanical pestle and processed for CUT&Tag analysis as previously described [104,105]. For normalizing the levels of histone H3 modifications and as a proxy of the starting material in each condition, we also generated samples for total histone H3. Libraries were prepared by supplementing the NEBNext HighFidelity 2× PCR Master Mix with 1 pg of Tn5-tagmented lambda DNA (New England Biolabs, #N3011S) as a spike-in normalizer. The library indexing was performed using Illumina i5 and i7 [106] through 15 cycles (1 × 5 min at 72 °C, 1 × 30 s at 98 °C, 13 × 10 s at 98 °C, 30 s at 63 °C, 1 × 1 min at 72 °C, hold at 4 °C). The libraries were purified using Nucleomag NGS Clean-up and Size Select beads (Macherey-Nagel, #744970.50). Quality checks were done with Qbit DNA HS Assay (ThermoScientific, #Q32854) and Bioanalyzer (Agilent). Finally, two independent library replicates were sent for next generation sequencing.

## Quantifications, bioinformatic analysis, and statistics

### General comments

For all quantifications of imaging data, control and experimental samples were processed together and imaged in parallel, using the same confocal settings. Images were processed, analyzed, and quantified using tools in Fiji (ImageJ 2.9.0)

[107]. Extreme care was taken to apply consistent methods (i.e., number of projected sections, thresholding methods, processing) for image analysis. Figure panels were assembled using Affinity Designer 2.6.2. Statistical analyses were performed in GraphPad Prism. Illustrations were created in BioRender.

## Quantification of histone modifications

For quantification of levels of histone modifications, an xy-representation containing the maximum number of pouch and hinge cells was generated using a sum projection of the relevant channel. Three square regions of interest (ROIs) measuring 15 × 15 µm were placed in distinct areas of either the pouch or the hinge, carefully excluding regions with developmentally arrested cells. The mean intensity of the modification signal was measured within each ROI. For each disc, the average pouch signal was normalized to the average hinge signal and reported as a pouch-to-hinge intensity ratio.

For Fig 3: An *xy* image with maximal coverage of anterior ZNCs, developmentally arrested, and dorsal senescence patches was generated by sum projection of the relevant channel. For ZNC quantification, two square ROIs (15 × 15 µm) were placed inside and outside the anterior ZNC region, and the mean intensity of modification signal was measured in each ROI. For each disc, the average "in-ZNC" intensity was normalized to the average "out-of-ZNC" intensity and reported as an in/out ratio. For developmentally arrested and dorsal senescence patches, a rectangular ROI encompassing the entire patch was drawn, and an identically sized ROI was placed immediately adjacent to the patch as a local background/neighbor reference. Mean modification intensity was measured within both ROIs, and patch enrichment was calculated as the ratio of patch to adjacent region for each disc.

## Quantification of EdU incorporation

S1 and S2 Fig: To first create a nuclear mask, an xy- representation containing the maximum number of pouch nuclei was generated using a sum projection of the DAPI channel. The background signal was reduced using the "Subtract Background" function with a rolling ball radius of 100 pixels. Local contrast was enhanced with the "Enhance Local Contrast (CLAHE)" function, using a block size of 127, 256 histogram bins, and a maximum slope of 3. Automatic thresholding was performed using the "Moments" method with the "dark" option selected, and the image was then converted into a binary mask of DAPI-positive nuclear areas. A nuclear ROI was generated from this mask using the selection tool. The pouch region was manually selected based on wing fold morphology. To isolate the nuclei within the pouch, a binary "AND" operation was used to combine nuclear mask and pouch selection. This final pouch-specific DAPI mask was used to measure EdU signal intensity.

S12 Fig: To create an EdU mask, an xy- representation containing the maximum number of pouch nuclei was generated using a sum projection of the EdU channel. A "Gaussian blur" ($\sigma = 2$) was applied to reduce noise and smooth the signal. Automatic thresholding was then performed using the "Moments" method with the "dark" option selected. The image was converted into a binary EdU mask with a black background. An ROI was generated from this mask using the selection tool to define EdU-positive pixels. Separately, the pouch region was manually selected based on wing fold morphology. To isolate EdU-positive pixels specifically within the pouch, a binary "AND" operation was applied to combine the EdU selection with the pouch selection. This final pouch-specific EdU mask was used to measure EdU signal intensity.

To assess changes in the number of proliferating cells, pouch-specific DAPI and EdU masks were generated as described above. A binary "AND" operation was used to extract EdU-positive nuclei located within DAPI-positive regions in the pouch. The area of EdU-positive nuclei was then divided by the total area of EdU and DAPI-positive regions within the pouch to calculate the fraction of proliferating cells. This EdU-to-DAPI area ratio was calculated per disc and used for comparative analysis.

## Quantification of cell cycle phase distribution in cell cycle manipulation models

To generate a nuclear mask, a single z-stack containing the maximum number of pouch cells was selected from DAPI channel. The background signal was reduced using the "Subtract Background" function with a rolling ball radius of 100 pixels. Local contrast was enhanced with the "Enhance Local Contrast (CLAHE)" function, using a block size of 127, 256 histogram bins, and a maximum slope of 3. Automatic thresholding was performed using the "Otsu" method with the "dark" option selected, and the image was then converted into a binary mask of DAPI-positive nuclear areas. A nuclear ROI was generated from this mask using the selection tool. The pouch region was manually selected based on wing fold morphology. To isolate the nuclei within the pouch, a binary "AND" operation was used to combine nuclear mask and pouch selection. This mask was overlaid on the GFP and RFP channels, and the X/Y coordinates of each ROI were used to extract signal intensities for each nuclear pixel. Threshold values for GFP and RFP determined based on average intensity of 5 nuclei in early S-phase. Threshold finding was based on how FUCCI intensities correlated with EdU incorporation patterns that distinguish euchromatic (early S-phase) from heterochromatic (late S-phase) DNA replication: FUCCI-negative cells show euchromatic EdU patterns (thus classified as early S-phase), whereas FUCCI-red show EdU enrichment in heterochromatic domains (consistent with late S-phase replication) (see S11A Fig). Each nuclear pixel was classified into a cell cycle phase using these thresholds, following FUCCI-based criteria: G2 phase: GFP > threshold, RFP > threshold; G1 phase: GFP > threshold, RFP ≤ threshold; Early/mid S-phase: GFP ≤ threshold, RFP ≤ threshold; Late S-phase: GFP ≤ threshold, RFP > threshold. Phase distributions were summarized as per-disc ratios.

## Quantification of OPP levels in wing imaginal discs

An xy-representation containing the maximum number of pouch cells was generated using a sum projection of the relevant channels. In both control and constitutively active insulin receptor-expressing discs, the pouch region was manually selected based on wing fold morphology. Mean OPP intensity was then measured within the defined pouch region for each disc.

## Quantification of pouch size

For both control and constitutively active insulin receptor-expressing discs, a slice containing the maximum number of pouch cells was selected. In each case, the pouch region was manually defined based on wing fold morphology. The area size of the selected pouch region was then measured for each disc.

## Cell cycle phase classification using FUCCI and quantifications of H3, Total Acetyl Lysine, H3K27ac, H3K18ac, and H4K8ac level

To classify nuclei in the wing pouch according to cell cycle phase and to quantitatively correlate levels of H3 and acetylation levels with each phase, a custom image analysis workflow was implemented in ImageJ/Fiji. A single z-slice containing the highest number of pouch cells was selected from each control disc. The image stack was split into individual channels (e.g., DAPI, GFP-E2f1[1-230], mRFP-NLS-CycB[1-266], and H3K27ac). To generate a nuclear mask, the DAPI, GFP, RFP, and other nuclear markers were averaged sequentially using the Image Calculator. The resulting image was thresholded using the Otsu method (upper limit set to 255) and converted into a binary mask defining nuclear pixels. A nuclear pixel selection was generated from the binary image and added to the ROI Manager. The pouch region was manually defined based on wing fold morphology. A binary "AND" operation was used to restrict the nuclear pixel mask to the pouch, producing a final pouch-specific nuclear mask. This mask was overlaid on the GFP, RFP, and other nuclear marker channels, and the X/Y coordinates of each ROI were used to extract signal intensities for each nuclear pixel. Threshold values for GFP and RFP determined based on average intensity of 5 nuclei in early S-phase. Each nuclear pixel was classified into a cell cycle phase using these thresholds, following FUCCI-based criteria: G2 phase: GFP > threshold, RFP > threshold;

G1 phase: GFP > threshold, RFP ≤ threshold; Early/mid S-phase: GFP ≤ threshold, RFP ≤ threshold; Late S-phase: GFP ≤ threshold, RFP > threshold. For each nuclear pixel, the intensity of the corresponding nuclear markers (e.g., H3, acetyl-lysine, and H3K27ac) was recorded. To account for inter-sample variability, intensity values within each cell cycle phase were normalized by dividing them by the total nuclear marker signal summed across all phases (G1, early S, late S, and G2) within the same disc. This normalization provided the relative contribution of each cell cycle phase to the total signal, allowing for direct comparisons across samples and avoid privileging any single phase as the normalization reference.

**CUT&Tag bioinformatic analysis**

The CUT&Tag data processing was performed as described before with some modifications [105,108,109]. As our quantitative CUT&Tag sequencing libraries contained DNA from *D. melanogaster* and spike-in of *lambda* phage genome (Genbank: J02459.1), we mapped the raw paired-end reads to a constructed hybrid dm6 and *Lambda* phage genome using *snakePipes-v3.0.0 DNA-mapping* [110] with updated Bowtie2 mapping parameters, as well as adapter trimming and MAPQ ≥ 1 filtering (*DNAmapping --mapq 1 --trim --fastqc --properPairs --dedup --cutntag --DAG --alignerOpts='--local --very-sensitive-local --no-discordant --no-mixed -I 10 -X 700'*). To assess reproducibility of both replicates, we utilized the *multiBamSummary* and *plotCorrelation* from *deepTools-v.2.5.7* to compute the Spearman correlation between the two replicates. Aligned replicates were merged before normalization to both H3 and spike-in signals. The code used for the H3 and *Lambda* normalization analysis was based on work done by Yinxiu Zhan (https://github.com/zhanyinx/atinbayeva_paper_2023) for [108]. CUT&Tag peaks were called using *snakePipes-v3.0.0 ChIPseq* with *MACS2-v 2.2.9.1* (*ChIPseq -d Output_spikein hybrid.yaml H3_spikein_config.yaml --peakCaller MACS2 --cutntag --peakCallerOptions "--broad --qvalue 0.5 -f BAMPE" --DAG*). Normalized bigWig and bam files were generated with *deepTools-v3.5.6* and used to compute matrices of signal enrichment ±5kb around transcription start sites (TSS) or peak centers of shared or unique peaks for *wt* and *Cdk1KD*. Coverage heatmaps and profiles were created using *plotHeatmap* or *plotProfile* from *deepTools-v.2.5.7*.

For the quantification, the number of PE reads was counted in each shared or unique peak for *wt* and *Cdk1KD* (*multiBamSummary BED-file --bamfiles <input files> --extendReads --outRawCounts <file>*) as well as in 500 bp bins across the dm6 genome (*multiBamSummary bins --bamfiles <input files> --binSize 500 --extendReads --outRawCounts <file>*). The *DESeq2-v1.38.3* analysis was performed using the previously computed H3 and spike-in scaling factors from merged data for size factor. Bins and peaks with basemean (average read count across samples) of less than or equal to 25 read counts were discarded. The label "no peaks called" represents only the 500 bp bins which do not include any sequence called by peak calling. The design matrix was setup to compare the samples by condition and correct for replicate effects (*design = ~replicate + condition*). Finally, we executed the *DESeq2* shrinkage of log2 fold changes (*type = 'normal'*).

## Supporting information

**S1 Table. Experimental strains, reagents, and software.** List of *Drosophila* strains, transgenes, antibodies, and software used in this study. Sources, providers and unique IDs are provided were available.
(DOCX)

**S1 File. Source Data and Statistics.** File containing individual sheets for each figure reporting data points and statistics for immunofluorescence experiments. The name of the sheet corresponds to the respective figure in the manuscript. A sheet stating number of fully controlled, stage-matched, independent biological replicates performed for each experiment, with at least 10–13 discs per genotype (except for rare genotypes), is included.
(XLSX)

**S1 Fig. Cell cycle manipulation models for assessing histone modification dynamics. A.** A control wing disc after 24 h of *UAS-GFP*-expression in the pouch, under the control of the *rn-GAL4* (*rotund-GAL4*) driver. GFP-expression visualizes the tissue domain subject to manipulation throughout this study (magenta). **B.** Schematic representation of the

Fly-FUCCI cell cycle reporter system, utilizing the degradable *GFP-E2F1*[1-230] (green) and *mRFP-NLS-CycB*[1-266] (red) as sensors to visualize cell cycle phases. Individual fluorophore expression indicates cells in G1 or late S-phase, respectively. Combined expression of both GFP and RFP labels cells is observed in early and late G2. Cells in late mitosis and early S-phase lack expression of either GFP or RFP. Cells in early S-phase are specifically detected by EdU incorporation [63,66]. Cells in mitosis are positioned in very apical positions of the tissue and are not represented in our study. **C–F.** Cell cycle dynamics in control discs (C, E) and discs expressing *Cdk1-RNAi* for 24 h under the control of *rn-GAL4* (D, F). FUCCI reporters *GFP-E2F1*[1-230] (green) and *mRFP-NLS-CycB*[1-266] (red) were used to visualize cell cycle phases (C, D) and EdU incorporation was used to detect DNA replication in S-phase cells (E, F). Cdk1 knockdown in the wing pouch domain leads to a cell cycle phase shift toward G2 phase (D), and a corresponding loss of EdU incorporation (F) combined demonstrating a pronounced arrest in G2. **G.** Quantified cell cycle phase ratios in control and *Cdk1-RNAi-*expressing discs (see Materials and Methods for details). Each dot represents one disc; symbols/colors denote the indicated genotypes and cell cycle phases. Error bars indicate mean ± SD. **H.** Quantification of mean EdU intensity per DAPI area in the pouch region of control and *Cdk1-RNAi-*expressing discs, serving as a proxy for relative DNA replication activity. Mean and 95% CI are shown. Statistical significance was tested using two-tailed Welch's test (control discs: $n = 11$, *Cdk1-RNAi-*expressing discs: $n = 12$). **I–L.** Cell cycle dynamics in control discs (I, K) and discs expressing *pntP1* (J, L) for 24 h under the control of *rn-GAL4.* FUCCI reporters *GFP-E2F1*[1-230] (green) and *mRFP-NLS-CycB*[1-266] (red) were used to visualize cell cycle phases (I, J) and EdU incorporation was used to detect DNA replication in S-phase cells (K, L). Pointed-P1 overexpression in the wing pouch domain leads to a cell cycle phase shift towards either the G1 or the G2 phase (J), correlating with a loss of EdU incorporation (L). Combined, this demonstrates a pronounced arrest in either G1 or G2. Mechanistically, the choice for either cell cycle phase is not known. **M.** Quantified cell cycle phase ratios in control and *pntP1*–expressing discs. Each dot represents one disc; symbols/colors denote the indicated genotypes and cell cycle phases. Error bars indicate mean ± SD. **N.** Quantification of mean EdU intensity per DAPI area in the pouch region of control and *pntP1*-expressing discs, serving as a proxy for relative DNA replication activity. Mean and 95% CI is shown. Statistical significance was tested using two-tailed Mann–Whitney test (control discs: $n = 6$, *pntP1*-expressing discs: $n = 6$). Discs were stained with DAPI to visualize nuclei. Fluorescence intensities are reported as arbitrary units. Yellow dashed lines represent the pouch region of wing discs. Sum projections of multiple confocal sections are shown in A, E and F, and K and L. Maximum projections of multiple confocal sections are shown in **C-D and I-J.** Scale bars: 100 μm. See S1 File for underlying data and statistical information.
(TIFF)

**S2 Fig. Cell cycle manipulation models for assessing histone modification dynamics. A–D.** Cell cycle dynamics in control discs **(A, C)** and discs expressing *Dp, E2F1* **(B, D)** for 24 h under the control of *rn-GAL4.* FUCCI reporters *GFP-E2F1*[1-230] (green) and *mRFP-NLS-CycB*[1-266] (red) were used to visualize cell cycle phases **(A, B)** and EdU incorporation was used to detect DNA replication in S-phase cells **(C, D).** dDp and dE2F1 co-expression in the wing pouch domain reduces the number of cells in either G1 or G2 **(E)**, and an increase in S-phase cells as well as an elevation of EdU incorporation **(E** and **F)** in the wing pouch region, confirming frequent entry and rapid progression through the cell cycle. **E.** Quantified cell cycle phase ratios in control and *Dp, E2F1*-coexpressing discs. Each dot represents one disc; symbols/colors denote the indicated genotypes and cell cycle phases. Error bars indicate mean ± SD. **F.** Quantification of mean EdU intensity per DAPI area in the pouch region of control and *Dp, E2F1*-coexpressing discs, serving as a proxy for relative DNA replication activity. Mean and 95% CI is shown. Statistical significance was tested using two-tailed Welch's *t* test (control discs: $n = 6$, *Dp, E2F1*-expressing discs: $n = 5$). **G-J.** Cell cycle dynamics in control discs **(G** and **I)** and discs expressing *CyclinE-RNAi* **(H)** or *dacapo* **(J)** for 24 h under the control of *rn-GAL4.* Cell cycle phases were visualized using the FUCCI reporters *GFP-E2F1*[1-230] (green) and *mRFP-NLS-CycB*[1-266] (red) **(G-J).** H3K27ac immunostaining was used to asses cell cycle-dependent changes in histone modifications. CyclinE knockdown and dacapo expression in the wing pouch shift cells toward G1 phase **(H** and **J)**, and are accompanied by reduced H3K27ac levels. Discs were stained with

DAPI to visualize nuclei. Fluorescence intensities are reported as arbitrary units. Yellow dashed lines represent the pouch region of wing discs. Sum projections of multiple confocal sections are shown in **A-B, C-D, G-H** and **I-J.** Scale bars: 100 µm. See S1 File for underlying data and statistical information.
(TIFF)

**S3 Fig. Histone modifications are largely insensitive to cell cycle manipulations. A–D.** Immunostaining for H3K9ac in control **(A)**, *Cdk1-RNAi*-expressing **(B)**, *pntP1*-expressing **(C)** and *Dp, E2F1*-expressing **(D)** discs. Expression was induced for 24 h in the wing pouch using *rn-GAL4*. The *rn-GAL4* expression pouch domain is represented by the tissue above the yellow line (rn$^{ts}$>). The tissue below the yellow line represents wild-type tissues of the hinge. Icons represent the experimentally verified FUCCI cell cycle status (see S1 and S2 Fig) in each condition: G2-phase arrest in *Cdk1-RNAi*-expressing discs **(B)**, combined G1 and G2-phase arrest in *pntP1*-expressing discs **(C)**, and cell cycle acceleration in *Dp, E2F1*-coexpressing discs **(D)**. **E-H**. Immunostaining for H4K8ac in control **(E)**, *Cdk1-RNAi*-expressing **(F)**, *pntP1*-expressing **(G)** and *Dp, E2F1*-coexpressing **(H)** discs. **I-L.** Immunostaining for total acetylated lysine in control **(I)**, *Cdk1-RNAi*-expressing **(J)**, *pntP1*-expressing **(K)** and *Dp, E2F1*-coexpressing **(L)** discs. **M-P.** Immunostaining for H3K18crotonylation in control **(M)**, *Cdk1-RNAi*-expressing **(N)**, *pntP1*-expressing **(O)** and *Dp, E2F1*-coexpressing **(P)** discs. **Q-T.** Immunostaining for total H3K9me1/2/3 methylation in control **(Q)**, *Cdk1-RNAi*-expressing **(R)**, *pntP1*-expressing **(S)** and *Dp, E2F1*-coexpressing **(T)** discs. **U.** Quantification of relative signal intensities for H3K9ac, H4K8ac, total acetylated lysine, H3K18crotonylation, and H3K9me1/2/3, presented as pouch-to-hinge ratios (rn$^{ts}$-to-wild-type-cell ratio) in *Cdk1-RNAi*-expressing discs. Mean and 95% CI is shown. Statistical significance was tested using two-tailed Unpaired *t* test for H3K9ac (control discs: *n* = 5, *Cdk1-RNAi*-expressing discs: *n* = 5); two-tailed Unpaired *t* test for H4K8ac (control discs: *n* = 5, *Cdk1-RNAi*-expressing discs: *n* = 5); two-tailed Unpaired *t* test for total acetylated lysine (control discs: *n* = 6, *Cdk1-RNAi*-expressing discs: *n* = 6); two-tailed Mann–Whitney test for H3K18crot (control discs: *n* = 11, *Cdk1-RNAi*-expressing discs: *n* = 11); two-tailed Unpaired *t* test for H3K9me1/2/3 (control discs: *n* = 9, *Cdk1-RNAi*-expressing discs: *n* = 10). **V.** Quantification of relative signal intensities for H3K9ac, H4K8ac, total acetylated lysine, H3K18crotonylation, and H3K9me1/2/3, presented as pouch-to-hinge ratios (rn$^{ts}$-to-wild-type-cell ratio) in *pntP1*-expressing discs. Mean and 95% CI is shown. Statistical significance was tested using two-tailed Unpaired *t* test for H3K9ac (control discs: *n* = 8, *pntP1*-expressing discs: *n* = 10); two-tailed Unpaired *t* test for H4K8ac (control discs: *n* = 12, *pntP1*-expressing discs: *n* = 12); two-tailed Unpaired *t* test for total acetylated lysine (control discs: *n* = 6, *pntP1*-expressing discs: *n* = 6); two-tailed Unpaired *t* test for H3K18crot (control discs: *n* = 7, *pntP1*-expressing discs: *n* = 7); two-tailed Mann–Whitney test for H3K9me1/2/3 (control discs: *n* = 6, *pntP1*-expressing discs: *n* = 6). **W.** Quantification of relative signal intensities for H3K9ac, H4K8ac, total acetylated lysine, H3K18crotonylation, and H3K9me1/2/3, presented as pouch-to-hinge ratios (rn$^{ts}$-to-wild-type-cell ratio) in *Dp, E2F1*-coexpressing discs. Mean and 95% CI is shown. Statistical significance was tested using two-tailed Unpaired *t* test for H3K9ac (control discs: *n* = 8, *Dp, E2F1*-coexpressing discs: *n* = 8); two-tailed Mann–Whitney test for H4K8ac (control discs: *n* = 11, *Dp, E2F1*-coexpressing discs: *n* = 11); two-tailed Mann–Whitney test for total acetylated lysine (control discs: *n* = 6, *Dp, E2F1*-coexpressing discs: *n* = 6); two-tailed Mann–Whitney test for H3K18crot (control discs: *n* = 7, *Dp, E2F1*-coexpressing discs: *n* = 7); two-tailed Unpaired *t* test for H3K9me1/2/3 (control discs: *n* = 10, *Dp, E2F1*-coexpressing discs: *n* = 10). Fluorescence intensities are reported as arbitrary units. Sum projections of multiple confocal sections are shown in **C-T.** Scale bars: 100 µm. Please refer to (S4B Fig) for corresponding DAPI images of all discs shown. See S1 File for underlying data and statistical information.
(TIFF)

**S4 Fig. Corresponding DAPI images for Figs 1 and S3.** Representation of DAPI images matched to the images of histone modifications in control, *Cdk1-RNAi*-expressing, *pntP1*-expressing and *Dp, E2F1*-coexpressing wing discs shown in Figs 1 and S3. Discs were stained with DAPI to visualize nuclei. Sum and max projections of multiple confocal sections are shown. Scale bars: 100 µm.
(TIFF)

**S5 Fig. Comparative analysis of Histone H3 and H3K27ac, H3K27me3, and H3K9me3 histone modifications by CUT&Tag in control and *Cdk1 RNAi-* expressing discs. A.** Schematic illustrating the genomic region categories used in the analysis. Each genomic region was categorized as containing peaks shared between WT and Cdk1-RNAi, peaks unique to WT, peaks unique to Cdk1-RNAi, or a 500-bp region with no called peaks in either condition. **B**. Table summarizes the total number of genomic regions described in **(A)** tested for each histone modification (H3, H3K27ac, H3K27me3, and H3K9me3) and the number of these regions that are significantly upregulated (log2FoldChange> 0) or significantly downregulated (log2FoldChange< 0) in Cdk1-RNAi relative to WT. **C-F.** MA plots displaying differential enrichment of histone modifications measured by CUT&Tag for H3, H3K27ac, H3K27me3, and H3K9me3 (rows). Each point represents a genomic region classified by peak category as defined in our analysis: regions with peaks shared between WT and Cdk1-RNAi **(C)**, peaks unique to WT **(D)**, peaks unique to Cdk1-RNAi **(E)**, or 500-bp regions with no called peaks in either condition **(F)**. The x-axis shows the mean normalized read count (log10), and the y-axis shows the log2 fold change (Cdk1-RNAi/WT). Points are colored by histone modifications (H3, blue; H3K27ac, orange; H3K27me3, green; H3K9me3, red). Regions with *DES*eq2 adjusted P< 0.1 are shown in color, whereas non-significant regions (ns; adjusted P ≥ 0.1) are shown in gray. Access to underlying data is provided at NCBI (SRA) PRJNA130380. See Code and Data Availability Statement.
(TIFF)

**S6 Fig. Cell cycle-dependent fluctuations of H3K27ac and H3K27me3 are not due to competitive residue occupancy. A-B.** H3K27me3 staining in control **(A)** or *E(z)-RNAi*-expressing **(B)** discs. Knockdown of the H3K27me3 writer E(z) causes a reduction in H3K27me3 level in the pouch of the wing disc. **C-D.** H3K27ac staining in control **(C)** or *E(z)-RNAi*-expressing **(D)** discs. Knockdown of the H3K27me writer E(z) causes no corresponding increase in H3K27ac level. **E-F.** H3K27ac staining in control **(E)** and *nejire-RNAi*-expressing **(F)** discs. Knockdown of the H3K27ac writer CBP/nej causes a reduction of H3K27ac level in the pouch of the wing disc. **G-H.** H3K27me3 staining in control **(G)** and *nejire-RNAi*-expressing **(H)** discs. Knockdown of the H3K27ac writer CBP/nej causes no corresponding increase in H3K27me3 level. Discs were stained with DAPI to visualize nuclei. Maximum projections of multiple confocal sections are shown in **A-B.** Sum projections of multiple confocal sections are shown in **C-D, E-F** and **G-H.** Scale bars: 100 µm.
(TIFF)

**S7 Fig. Visualization of cell cycle-arrested cell populations and histone modifications in developing wing imaginal disc. A.** Schematic representation of the zone of nonproliferation cells (ZNC) (adapted from [72]). **B-B'.** The Fly-FUCCI system was used to visualize the spatial pattern of developmentally arrested cells (cyan arrowhead) and dorsal senescence cells (yellow arrowhead) in hinge region of normally developing wing disc. **C-D.** Immunostaining for H3K18ac **(C)** and H3K4me3 **(D)** in the zone of nonproliferation cells (ZNC) of developing wing discs. Neither histone modification shows detectable changes in this region. **E-F.** Immunostaining for H3K18ac **(E)** and H3K4me3 **(F)** in the wing disc hinge region; programmed senescence is indicated by yellow arrowheads and developmental cell cycle arrest is indicated by cyan arrowheads. **G.** Quantification of ratios of histone modification levels within the anterior ZNC, developmentally arrested cells, and dorsal senescence cells relative to regions outside each domain (see Materials and Methods for details) in normally developing wing discs. For each condition, a one-sample Wilcoxon signed-rank test was used to test whether the median ratio differed from 1. Symbols represent individual wing discs; horizontal black lines indicate the median and error bars indicate the IQR. For H3K4me3; anterior ZNC $n = 15$, developmentally arrested cells $n = 11$, dorsal senescence cells $n = 11$. For H3K27me3; anterior ZNC $n = 8$, developmentally arrested cells $n = 8$, dorsal senescence cells $n = 8$. For H3K9me3; anterior ZNC $n = 10$, developmentally arrested cells $n = 10$ and dorsal senescence cells $n = 10$. For H3K18ac; anterior ZNC $n = 13$, developmentally arrested cells $n = 13$ and dorsal senescence cells $n = 13$. For H3K27ac; anterior ZNC $n = 11$, developmentally arrested cells $n = 11$ and dorsal senescence cells $n = 11$. Discs were stained with DAPI to visualize

nuclei. Fluorescence intensities are reported as arbitrary units. Sum projections of multiple confocal sections are shown in **B-B'**. Scale bars: 100 μm in **B**; 50 μm in **B'** and **C-H**. See S1 File for underlying data and statistical information.
(TIFF)

**S8 Fig. Polycomb target gene expression remains stable despite altered cell cycle dynamics. A-L**. Immunostaining for proteins expressed by Polycomb target genes: Cut **(A, B)**, Engrailed/invected **(C, D)**, Nubbin **(E, F)**, Antennapedia **(G, H)**, Patched **(I, J)** and Wingless **(K, L)** in control **(A, C, E, G, I, K)** and *Cdk1-RNAi*-expressing **(B, D, F, H, J, L)** wing discs. **M-R.** Immunostaining for proteins expressed by Polycomb target genes: Antennapedia **(M, N)**, Engrailed/invected **(O, P)** and Patched **(Q, R)** in control **(M, O, Q)** and *Dp, E2F1*-coexpressing **(N, P, R)** wing discs. **S.** A control wing disc after 24 h of *UAS-GFP*-expression in the pouch (magenta), under the control of the *rn-GAL4* (*rotund-GAL4*) driver, displayed to reference the manipulated pouch domain in Figs A-R. Please note that this is the same disc shown in S1A Fig. Discs were stained with DAPI to visualize nuclei. Sum projections of multiple confocal sections are shown in **A-S**. Scale bars: 100 μm.
(TIFF)

**S9 Fig. Increased insulin signaling promotes faster proliferation and growth independent of changes to the distribution of cell cycle phases. A-B.** Protein synthesis visualized by OPP incorporation in control **(A)** and *InR-DA*-expressing **(B)** discs. Yellow dashed lines indicate the boundary between the wing pouch and hinge regions. The area above the line corresponds to the *rn-GAL4* expression domain, while the area below represents wild-type cells. The *InR-DA*-expressing wing pouch exhibits higher rates of protein synthesis, reflecting an increased biosynthetic capacity promoted by higher InR/PI3K/Akt activity. **C.** Quantification of mean OPP intensity in the pouch region of control and *InR-DA*-expressing discs. Mean and 95% CI is shown. Statistical significance was tested using two-tailed Unpaired *t* test (control discs: $n=5$, *InR-DA*-expressing discs: $n=6$). **D-E.** EdU incorporation visualizes DNA replication in the pouch region of control **(D)** and *InR-DA*-expressing **(E)** discs. Dashed yellow squares highlight the magnified regions shown in **(D'** and **E')**. *InR-DA*-expressing wing pouch shows higher levels of EdU intensities, reflecting accelerated speed of DNA replication. Rates of EdU incorporation in *InR-DA*-expressing discs were previously reported and quantified [63]. **F-G.** Cell cycle dynamics in control **(F)** and *InR-DA*-expressing **(G)** discs. FUCCI reporters *GFP-E2F1*[1-230] (green) and *mRFP-NLS-CycB*[1-266] (red) were used to visualize cell cycle phases. Note that expression of a constitutively active Insulin receptor does not alter the cell cycle phase profile of the wing pouch compared to control disc. **H.** Illustration of the dual role of metabolism in regulating H3K9 and H3K27 trimethylation levels. Metabolism contributes to the dilution of histone PTMs by accelerating the cell cycle, thereby increasing the frequency of DNA replication events. Simultaneously, it supports the reestablishment of these PTMs by supplying essential precursors and cofactors required for methyltransferase activity. **I.** Hypothetical model depicting the relationship between histone H3K9 and H3K27 trimethylation levels, metabolic rate, and cell division rate. When metabolic activity and cell division are proportionally coupled (black diagonal), H3K9 and H3K27 trimethylation levels are maintained at stable levels. However, when the cell cycle is slowed or when metabolism outpaces cell division, H3K9 and H3K27 trimethylation accumulates. Discs were stained with DAPI to visualize nuclei. Fluorescence intensities are reported as arbitrary units. Maximum projections of multiple confocal sections are shown in **F-G**. Sum projections of multiple confocal sections are shown **A-B** and **D-E**. Scale bars: 100 μm in **A-B**, **D-E** and **F-G**; 50 μm in zoomed-in **D'** and **E'**. See S1 File for underlying data and statistical information.
(TIFF)

**S10 Fig. Persistence methyltransferase activity drives H3K9me3 accumulation through metabolic changes coupled to a senescent cell cycle arrest. A-B**. Senescence-associated β-galactosidase (SA-β-gal) activity (cyan or gray) in control **(A)** and *egr*-expressing **(B)** discs. TRE-RFP reporter (magenta) visualizes JNK-pathway activity. SA-β-gal activity in *egr*-expressing discs was previously reported and quantified [77]. **C-D.** EdU incorporation (gray or cyan) visualizes DNA replication in control **(C)** and *egr*-expressing **(D)** discs. TRE-RFP reporter (magenta) visualizes JNK-pathway activity. EdU incorporation was previously reported and quantified [77,76]. **E-F.** Cell cycle dynamics in control **(E)** and *egr*-expressing

**(F)** discs. FUCCI reporters *GFP-E2F1*[1-230] (green) and *mRFP-NLS-CycB*[1-266] (red) were used to visualize cell cycle phases. **G.** A model illustrating inflammatory tissue damage in *Drosophila* wing imaginal discs. Eiger expression activates the JNK pathway, an early stress response, which induces a senescence-like cell cycle arrest in the G2 phase at center of the tissue damage and triggers a senescence program [64]. These arrested cells secrete Unpaired cytokines, which activate the JAK/STAT pathway in surrounding cells, promoting compensatory proliferation during tissue regeneration [63,76,77,111–114]. At the periphery of the damage site, cells remain in a quiescent state [115]. **H-I'.** Immunostaining for H3K9me3 in *egr*-expressing disc **(H)**. The magenta frame highlights G2-arrested senescent cells, which show elevated H3K9me3 levels. The cyan frame marks cells undergoing compensatory proliferation, where H3K9me3 levels are reduced. The yellow frame indicates quiescent cells, which have higher H3K9me3 levels than proliferating cells **(I)**. TRE-RFP reporter signal (magenta) is elevated in G2-arrested senescent cells but is absent in cells undergoing compensatory proliferation or quiescence **(I')**. Discs were stained with DAPI to visualize nuclei. Maximum projections of multiple confocal sections are shown in **H**. Sum projections of multiple confocal sections are shown in **A-B**, **C-D** and **E-F**. Scale bars: 100 µm in **A-B**, **C-D**, **E-F** and **H**; 10 µm in **I**.
(TIFF)

**S11 Fig. Total H3 and acetyl-lysine dynamics only partially recapitulate the S-phase-associated increase in H3K27ac. A.** FUCCI reporter and EdU incorporation assays in the peripodium of a wild-type wing disc. Euchromatic EdU incorporation (early S-phase) correlates with absence of fluorescence from both FUCCI reporters (*GFP-E2F1*[1-230] (green) and *mRFP-NLS-CycB*[1-266] (red); filled arrowheads). Heterochromatic EdU incorporation (late S-phase) correlates with a modest increase in the G2-specific reporter *mRFP-NLS-CycB*[1-266] (red; open arrowheads). Cells with elevated levels of both FUCCI reporters (yellow) are in late G2 [66]. **B-C.** Immunostaining for H3 in a developing control disc **(B)**, with a magnified region (demarcated by a white dashed square) shown in panel **C**, also expressing the FUCCI reporter (*GFP-E2F1*[1-230] (green) and *mRFP-NLS-CycB*[1-266] (red)) to visualize cell cycle phases. **D.** Quantification of normalized H3 intensity level with respect to different cell cycle phases in the pouch region of control discs. Normalization was performed to the average fluorescence intensity across all FUCCI-defined phases (to avoid privileging any single phase as the normalization reference). Mean and 95% CI is shown. Statistical significance was tested using Repeated Measures One-Way ANOVA followed by Tukey's post-hoc test for multiple comparison ($n = 11$ discs). **E-F.** Immunostaining for total acetylated lysine level in a developing control disc **(E)**, with a magnified region (demarcated by a white dashed square) in panel **F**, also expressing the FUCCI reporters (*GFP-E2F1*[1-230] (green) and *mRFP-NLS-CycB*[1-266] (red)) to visualize cell cycle phases. **G-H.** Immunostaining for H3K27ac in control **(G)** and *nejire*-expressing **(H)** discs with FUCCI reporters (*GFP-E2F1*[1-230] (green) and *mRFP-NLS-CycB*[1-266] (red)) to visualize cell cycle phases. The location of magnified regions shown in Main Fig 5I and 5J is indicated by white dashed squares. **I-J.** Immunostaining for H3K27ac in control **(I)** and *HDAC1-RNAi*-expressing **(J)** discs with FUCCI reporters (*GFP-E2F1*[1-230] (green) and *mRFP-NLS-CycB*[1-266] (red)) to visualize cell cycle phases. The location of magnified regions shown in Main Fig 5K and 5L is indicated by white dashed squares. **K-L.** Immunostaining for H3K27ac in control **(K)** and *HDAC1*-expressing **(L)** discs with FUCCI reporters (*GFP-E2F1*[1-230] (green) and *mRFP-NLS-CycB*[1-266] (red)) to visualize cell cycle phases. The location of magnified regions shown in Main Fig 5M and 5N is indicated by white dashed squares. Fluorescence intensities are reported as arbitrary units. Maximum projections of multiple confocal sections are shown in **E-F**. Scale bars: 100 µm. See S1 File for underlying data and statistical information.
(TIFF)

**S12 Fig. H3K18ac and H4K8ac dynamics only partially recapitulate the S-phase-associated increase in H3K27ac. A-B.** Immunostaining for H3K18ac in a developing control disc **(A)**, with a magnified region (demarcated by a white dashed square) in panel **B**, in the background of FUCCI reporters (*GFP-E2F1*[1-230] (green) and *mRFP-NLS-CycB*[1-266] (red)) to visualize cell cycle phases. **C.** Quantification of normalized H3K18ac intensity level in different cell cycle phases in the

pouch region of control discs. Normalization was performed to the average fluorescence intensity across all FUCCI-defined phases (to avoid privileging any single phase as the normalization reference). Mean and 95% CI is shown. Statistical significance was tested using Repeated Measures One-Way ANOVA followed by Tukey's post-hoc test for multiple comparison (*n* = 13 discs). **D-E**. Immunostaining for H4K8ac in a normally developing control disc **(D)**, with a magnified region (demarcated by a white dashed square) in panel **E,** also expressing the FUCCI reporters (*GFP-E2F1*[1-230] (green) and *mRFP-NLS-CycB*[1-266] (red)) to visualize cell cycle phases. **F.** Quantification of normalized H4K8ac intensity level in different cell cycle phases in the pouch region of control discs. Normalization was performed to the average fluorescence intensity across all FUCCI-defined phases (to avoid privileging any single phase as the normalization reference). Mean and 95% CI is shown. Statistical significance was tested using Repeated Measures One-Way ANOVA followed by Tukey's post-hoc test for multiple comparison (*n* = 12 discs). **G-H.** Immunostaining for H3K27ac in control **(G)** and *nejire-RNAi*-expressing **(H)** discs. Same disc is also shown in Fig 5E and 5F but repeated here to allow for direct comparison with **(I-N)**. **I-J.** Immunostaining for H3K18ac in control **(I)** and *nejire-RNAi*-expressing **(J)** discs. **K-L.** Immunostaining for H4K8ac in control **(K)** and *nejire-RNAi*-expressing **(L)** discs. **M-N.** Immunostaining for H3K18crot in control **(M)** and *nejire-RNAi*-expressing **(N)** discs. Yellow dashed lines indicate the boundary between the wing pouch and hinge regions. **O.** Expression of a Crisper-tagged CBP/nej-GFP [82] in a developing disc. **P-Q.** EdU incorporation to visualize DNA replication in the pouch region of control **(P)** and *nejire-RNAi*-expressing **(Q)** discs. Dashed yellow squares demarcate magnified regions **(P'**and **Q')**. Expression was induced for 24 h in the wing pouch using *rn-GAL4.* **R.** Quantification of mean EdU intensity per EdU area in the pouch region of control and *nejire-RNAi*-expressing discs, serving as a proxy speed of DNA replication. Mean and 95% CI are shown. Statistical significance was tested using two-tailed Mann–Whitney test (control discs: *n* = 11, *nejire-RNAi*-expressing discs: *n* = 11). **R'.** Quantification of EdU area per DAPI area in the pouch region of control and *nejire-RNAi*-expressing discs, serving as a proxy number of cells undergoing DNA replication. Mean and 95% CI are shown. Statistical significance was tested using two-tailed Mann–Whitney test (control discs: *n* = 11, *nejire-RNAi*-expressing discs: *n* = 11). Discs were stained with DAPI to visualize nuclei. Fluorescence intensities are reported as arbitrary units. Maximum projections of multiple confocal sections are shown in **A-B** and **D-E.** Sum projections of multiple confocal sections are shown in **G-H, I-J, K-L, M-N** and **P-Q.** Scale bars: 100 μm. See S1 File for underlying data and statistical information.
(TIFF)

## Acknowledgments

We thank the staff of the Life Imaging Center (LIC) in the Hilde Mangold House (HMH) of the Albert-Ludwigs-University of Freiburg for help with their confocal microscopy resources, and the excellent support in image recording. We specifically thank the DFG for supporting our imaging work through project number 414136422. We thank the Bioinformatics and Sequencing facilities at the MPI-IE for supporting this work. We thank Galina Erikson for supporting our bioinformatic analysis. We thank Laura Bauttita, Melissa Harrison, Iswar Hariharan, Dirk Bohmann, Carlos Estella, and Jürg Müller for sharing reagents and the Bloomington *Drosophila* Stock Center (BDSC), the Vienna *Drosophila* Stock Collection (VDRC), the University of Zurich ORFeome Project (FlyORF), and the Developmental Studies Hybridoma Bank (DSHB) for providing fly stocks and antibodies. We thank the IMPRS-EBM and SGBM graduate schools for supporting our doctoral researchers.

## Author contributions

**Conceptualization:** Liyne Nogay, Anne-Kathrin Classen.

**Data curation:** Liyne Nogay, Lara Heckmann, Anne-Kathrin Classen.

**Formal analysis:** Liyne Nogay, Lara Heckmann, Anne-Kathrin Classen.

**Funding acquisition:** Anne-Kathrin Classen.

**Investigation:** Liyne Nogay, Ananthakrishnan Vijayakumar Maya.

**Methodology:** Liyne Nogay, Ananthakrishnan Vijayakumar Maya, Lara Heckmann, Francesco Cardamone, Isabelle Grass, Aakriti Singh, Anna Frey, Laurin Ernst, Anne-Kathrin Classen.

**Project administration:** Isabelle Grass, Anne-Kathrin Classen.

**Resources:** Nicola Iovino, Anne-Kathrin Classen.

**Software:** Lara Heckmann.

**Supervision:** Nicola Iovino, Anne-Kathrin Classen.

**Validation:** Liyne Nogay, Ananthakrishnan Vijayakumar Maya, Francesco Cardamone, Isabelle Grass.

**Visualization:** Liyne Nogay, Lara Heckmann, Anne-Kathrin Classen.

**Writing – original draft:** Liyne Nogay, Anne-Kathrin Classen.

**Writing – review & editing:** Liyne Nogay, Ananthakrishnan Vijayakumar Maya, Lara Heckmann, Anne-Kathrin Classen.

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
