## [Editor Report · Decision Letter 0]

13 Aug 2025

Dear Dr Classen,

Thank you for submitting your manuscript entitled "Cell cycle dynamics regulate H3K27 and H3K9 histone modifications" for consideration as a Research Article by PLOS Biology.

Your manuscript has now been evaluated by the PLOS Biology editorial staff, as well as by an academic editor with relevant expertise, and I am writing to let you know that we would like to send your submission out for external peer review.

Once your full submission is complete, your paper will undergo a series of checks in preparation for peer review. After your manuscript has passed the checks it will be sent out for review. To provide the metadata for your submission, please Login to Editorial Manager (https://www.editorialmanager.com/pbiology) within two working days, i.e. by Aug 15 2025 11:59PM.

Kind regards,

Richard

Richard Hodge, PhD

rhodge@plos.org

PLOS

---

## [Decision Letter · Decision Letter 1]

16 Sep 2025

Dear Dr Classen,

Thank you for your patience while your manuscript "Cell cycle dynamics regulate H3K27 and H3K9 histone modifications" was peer-reviewed at PLOS Biology. Please accept my sincere apologies for the delays that you have experienced during the peer review process. Your manuscript has now been evaluated by the PLOS Biology editors, an Academic Editor with relevant expertise, and by four independent reviewers.

In light of the reviews, which you will find at the end of this email, we would like to invite you to revise the work to thoroughly address the reviewers' reports.

As you will see, the reviewers are generally positive and think the findings are interesting. However, they raise overlapping concerns about the lack of quantitative and statistical analyses for several experiments, including the FUCCI, CUT&Tag and immunofluorescence assays. In addition, Reviewer #1 raises concerns about the interpretation of the findings with regards to the effects on DNA replication stress. The reviewer notes that additional direct data should be provided to support the claims or that the data in Figure 6 is removed. After discussions with the Academic Editor, we agree with this comment and think that a revised version with Figure 6 removed would still be suitable for PLOS Biology.

Given the extent of revision needed, we cannot make a decision about publication until we have seen the revised manuscript and your response to the reviewers' comments. Your revised manuscript is likely to be sent for further evaluation by all or a subset of the reviewers.

**IMPORTANT - SUBMITTING YOUR REVISION**

*Re-submission Checklist*

*Published Peer Review*

*PLOS Data Policy*

*Blot and Gel Data Policy*

Best regards,

Richard

Richard Hodge, PhD

rhodge@plos.org

REVIEWS:

Reviewer #1: The manuscript by Nogay and coworkers addresses the interplay between cell cycle progression and histone modifications, which is an interesting and understudied area in the field of epigenetics. As a model system they use the Drosophila wing imaginal disc, which to my knowledge is the first attempt to study this in an in vivo system. By manipulating the cell cycle, the authors identify changes in H3K27me3, H3K9me3 and H3K27ac. They find that H3K27me3 and H3K9me3 accumulate in cells arrested in either G2 or G1, whereas E2F over-expression which should speed up the cell cycle leads to a reduction in these marks. H3K27ac show the opposite behavior.

They show that a physiological change of cell cycle speed would not lead to a change in H3K27me3 and H3K9me3 levels (what about H3K27ac?), and provide an attractive hypothesis that a balanced increase in proliferation and metabolic activity ensures that enzymatic activity matches the shorter cycling time.

They study H3K27ac separately and show that the levels are elevated in early S phase. They observed increased DNA damage upon depletion of the HAT Nejire and in a H3K27R line, and propose that H3K27ac directly counteracts replication stress.

The study is interesting and broadly relevant to the chromatin and cell cycle fields, in particular with respect to the findings on H3K27me3 and H3K9me3 which are exciting. However, the conclusions about the function of H3K27ac in replication stress protection are not convincingly supported by the data.

Concerns

1. The interpretation of the data on H3K27ac in Figure 5 and 6 is problematic.

Fig 5:

The authors first conclude that K27ac is higher in early and late S phase. Yet compared to G1, only early S phase is higher in K27ac. It is not advisable to compare to G2, as it is well established that the epigenome is deacetylated prior to entry into mitosis. Therefor the data suggest that early S phase cells are higher in K27ac, not S phase in general as stated.

The rational from the over-expression experiment is unclear. If you overexpress an enzyme, one may not expect that it maintains it normal regulation. Why is G2 and early S cells not shown?

For the HDAC1 knockdown, might the effect be related to the de-acetylation happening before mitosis? What happens in early S and G2?

Fig. 6:

The conclusion that CBP/nej is specifically required to prevent genome-wide replication stress cannot be made from the presented data. The rationale is flawed as it ignores the established role of K27ac in transcription.

Firstly, they provide no direct link between the DNA damage signal and S phase/ DNA replication. Would lack of replication prevent damage?

Secondly, damage could arise as a secondary effect of K27ac affecting expression of key replication/repair factors and/or replication-transcription conflicts. Thus, it cannot be concluded that the effect is direct. Direct effects on DNA replication / genome stability by K27ac would have to be established via specific mechanisms.

I therefore strongly recommend that this part is removed (Fig. 6).

2. The cell cycle distributions are not very well documented across the manuscript. The authors use the FUCCI system, but do not quantify the changes. This should be done across the different setups (Figure 1 and 4).

Over-expression of E2F1 is expected to shorten G1 and push cells faster into S phase, it is therefore surprising that the authors state that there are no changes in the proportion of G1 and G2 cells. It looks like there are more cells in S - which should lead to a proportional reduction in G1/G2. Quantification of the different cell cycle stages in the models should clarify this.

3. Please comment on whether the Cut&Tag analysis is quantitative. Was spike-in applied, and, if not, what caveats should be considered.

4. Overall the analysis of the Cut&Tag data appears superficial. It would be useful to expand on this and compare the number of peaks that go up versus down, and the intensity of these peaks in box plots or similar.

What is meant by 'other genomic regions'? was there a different threshold for this analysis? It seems that for both K9me3 and K27me3 the basal level across the genome is increased - outside peaks. Is this correct?

The H3K9me3 snapshot show rather limited changes compared to the IF results and H3K27me3 - can the authors comment? Would it be relevant to compare the behavior across repetitive regions versus repressed promoters?

Please also include annotations on the axis of all plots with genomics data.

5. The paper would benefit substantially from being more quantitative. All key conclusions from IF data should be supported by quantification across replicates and statistics. This is relevant for several figures (Fig 3, Fig 4,..) .

Please also make sure all IF panels are fully annotated. It would also be helpful if relevant cells were marked by arrows or circles on more of the images.

6. The conclusions regarding gene expression are also based on IF patterns that are not quantified. This does not exclude changes in gene expression, especially more subtle and heterogeneous ones that would not lead to strong changes in protein levels. Quantitative data should be provided, preferentially on the mRNA level.

7. The results on H3K27me3 are consistent with earlier findings that H3K27me3 accumulate in G1 /G0 arrested cultured cancer cells (Alabert et al., 2015) and a recent more elaborate study on H3K27me3 in ESCs (Trouth et al., 2025). Please include a discussion on this: the new work is in vivo and in a different organisms so discussing these findings together will help establish the general principles.

H3K9me3 was not found to accumulate in G1/G0 arrested cells (Alabert et al., 2015), but rather in S phase arrested cells experiencing replication stress (Gaggioli et al., 2023). Please also discuss these differences.

In relation to H3K9me3 regulation, the authors could also consider discussing DOI: 10.1126/science.adq7408

8. In the abstract first line, do the authors mean epigenome integrity rather than genome integrity?

9. The authors include a speculation on senescence in the results section as well as in the discussion. It should be removed from the results.

Reviewer #2: Nogay et al. show that levels of some histone modifications, but not others, are linked to the cell cycle in Drosophila wing imaginal discs. H3K9me3 and H3K27me3 accumulate as the cell cycle lengthens, and adopt to insulin signalling. By contrast, H3K27ac levels are highest in early S-phase and decline with longer cell cycles. Knock-down of the H3K27 acetylase CBP results in G2 arrest, and phospho-H2Avariant (gammaH2Av) accumulates in H3K27R mutant clones, indicative of replication stress upon H3K27ac reduction.

Overall, the paper presents a series of elegantly designed and well-explained experiments relying mainly on immunofluorescence in a Fly-FUCCI background to reveal a link between cell cycle and some histone modifications. The data are solid and will be of high interest to a wide community of biologists.

Comments

1. It is very intriguing that only some histone modifications are sensitive to perturbation of the cell cycle. In particular, H3K27ac is sensitive whereas H3K18ac and H4K8ac that are acetylated by the same enzyme (CBP/nejire) are not. Can the authors speculate on the reasons for this?

2. Expression of tested Polycomb-targets are not affected in Cdk1i or by Dp, E2F1 overexpression. Is this related to a block in cell cycle progression or that levels of H3K27me3 do not change enough? I would like to see a quantification of the H3K27me3 signal in the E(z)i experiment to be able to compare it with the Dp, E2F1 over-expression. Is the reduction in H3K27me3 achieved after 24h E(z)i sufficient to cause Polycomb-target de-repression?

3. On a similar note, the Cavalli lab showed that JAK-STAT signalling and its target ZFH1 become irreversibly up-regulated upon transient knock-down of the PRC1 component Ph (PMID: 38658752). Is this pathway affected upon cell-cycle perturbation?

4. If insulin signalling affects the abundance of metabolites such as SAM, why is not H3K4me3 levels altered? Metabolism should also affect the amount of acetyl-CoA, so how does H3K27ac look like?

5. Levels of total H3 and all acetylations are elevated in early S phase (Fig. S5.1 and 2). How come then that only H3K27ac is increased in cells with fast cell cycles?

6. What is known about how HDAC1 is regulated during the cell cycle? The previous literature on the effects of HDAC1 knockdown on the cell cycle should be discussed.

7. The EdU signal (Fig. S4.1E,D) in InR-DA and gammaH2Av levels in H3K27R mutant clones (Fig. 6K) should be quantified.

Minor comments

1. Line 162: there are two Buttitta et al references from the same year.

2. In Fig. S2.1, the counts for the unique peaks are very similar to "other genomic regions", questioning if they are really new peaks or just background signal.

3. The nuclei in inflammatory senescent cells look different from peripheral quiescent cells (Fig. S4.2). Can the authors be sure that higher H3K9me3 signal is not simply due to this difference rather than higher total levels of H3K9me3 in these cells?

4. Does nejire OE cause an increase in wing disc size or is the size difference in Fig. S5.1G-J due to rearing conditions?

5. Is cell cycle phase distribution affected by nejire OE?

6. Please comment on the difference between Fig. 5J and 5J', is there a bigger effect at other cell cycle stages?

7. What is the evidence for the statement on lines 372-373: "which is consistent with the reduced CBP/nej substrate specificity for these two residues". Please give references.

8. Can the signal in Fig.6F,G also be normalized on the level in the hinge region? The difference between hinge and pouch is hard to appreciate.

Reviewer #3: PLoS Biology Review: In this manuscript, the authors utilize fly genetics and the developing wing imaginal disc to characterize cell cycle associated changes in a set of histone post-translational modifications (hPTMs), suggesting roles for cell cycle associated chromatin dynamics in metabolism, senescence, and genome stability. The manuscript is well written and with interesting hypothesis, however, in its current state the analyses presented do not adequately and robustly support the author's conclusions. In many cases, proper quantification with statistics are missing.

Major Critiques:

* The authors provide a sophisticated repertoire of quantification methodologies that, when used, support their images and writing effectively. Unfortunately, these methodologies are used inconsistently and sparingly throughout this manuscript, with many conclusions being made qualitatively from individual images. For example, the primary GAL4 driver used in this manuscript, rn-GAL4, is expressed exclusively in the wing pouch, data from which can be normalized to the wing hinge, which acts as an internal control that the authors utilized in Figures 1U-W. This type of quantification is largely absent from the rest of the manuscript. If present, these qualifications would provide insight into the visually subtle changes in hPTMs described by the authors and would likely provide support for their conclusions.

* Given the spatial specificity of rn-Gal4, would be better to generate real mosaic clones using ubiquitous Gal4 drivers to compare RNAi or overexpression clones with adjacent controls. This strategy is particular powerful when using immunostaining signals to quantify and draw conclusions.

* Another example is the normalization of hPTMs, which in this work are mainly on H3. Therefore, normalization of each hPTMS by H3 signals would be more informative. In fact, the H3 by itself should also reflect cell cycle dynamics, as noted by authors in 351-353: "Of note, fluctuation of H3 and total lysine acetylation in our microscopy assays can be mostly explained by S-phase dependent histone synthesis and histone acetylation, as well as fluctuating morphology of interphase nuclei".

* Similarly, the authors use FUCCI to distinguish cell cycle phases in their analysis of hPTMs. The authors use a thresholding-based method for classifying cell cycle phase, but are inconsistent in providing quantification, often providing only large, heterogenous fields of view like seen in Figures 3A-C. This quantification is also necessary for all FUCCI data.

* Distinction between early and late S-phase appears to be arbitrary based on the author's thresholding approach. As late S-phase is associated with replication of constitutive heterochromatin, validation that these thresholds can distinguish between early and late S-phase can be achieved via EdU incorporation with anti-H3K9me3 antibody staining, or other image-based methods to distinguish early vs. late S-phase nuclei.

Other Critiques:

* Labels on microscopy images are very hard to read and sometimes missing.

* Fig 2A-B: When comparing the WT and Cdk1-RNAi histograms the change in peak height is not always obvious, which makes the "change" histograms look exaggerated. What may help is having a dotted line on the histograms at the WT peak maxima, which will act as a ruler in each histogram for differences in peak height. Also, where are the statistics of these comparisons?

* Fig 3: The locations of A-C and D-F on the wing disc are unclear, especially to a non-drosophilist. A diagram of the wing disc with these regions highlighted would be helpful for the reader. Again, where are the quantification and statistics of these data?

* Fig S4D-E: The authors state that there is "high EdU incorporation" in the InR-DA expressing wing pouch, however, visually the EdU in this tissue looks identical if not in fewer cells compared to the control. The figure legend states that this is based on high EdU intensity, however, this could result from different Click-iT reaction efficiencies. More appropriate analyses would be a S-phase index and/or colocalization analyses with between EdU and DAPI. Regardless, quantification is needed.

* Fig 4E: The authors state that there is no change in H3K27me3 compared to the control (4D), but a region of the wing pouch next to the dotted line looks like it has lower H3K27me3 than the control. Quantification would provide clarity on this.

* Fig 4M: The authors state that H3K4me3 does not change between 4L-O, however, it looks lower in 4M. Quantification would clarify this.

* Fig S4.2H: It is unclear if there is overlap between the inflamed region and the cyan "proliferating" region. SA-β-Gal or TRE-RFP included in these images would clear this up.

* Figure 6F-G: The authors report that upon nej knock-down there is an increase in γH2Av signal within the wing pouch, lending support to the hypothesis that nej protects the genome from replication stress induced DNA damage. Though γH2Av signal does appear more uniform in the nej RNAi wing pouch, it is unclear as to whether the signal within wing pouch nuclei is in fact greater than the control or even the wing hing of the same wing disc. Quantification is necessary to support this conclusion.

Reviewer #4: The manuscript entitled "Cell cycle dynamics regulate H3K27and H3K9 modifications" uses the wing imaginal disc to assess the overall connection between cell cycle progression and changes to the chromatin landscape. Using a range of physiological and altered conditions, the authors note associations between cell cycle phases and overall levels of H3K9me3, H3K27me3, and H3K27ac. At least one modification, H3K4me3, remained unchanged. These changes occur at distinct times during the cell cycle and appear to be regulated separately. In addition to carrying out staining to examine the "big picture" changes to chromatin that occur, the distribution of H3K27me3 and H3K9me3 were examined in more detail using CUT&TAG. Overall, the data described in this manuscript provide in vivo insights into the dynamic and complex relationship that must exist to allow for proliferation and genomic fidelity.

Several parts of the manuscript could be clearer or expanded upon to strengthen the story.

1. In Figure 2, CUT&Tag data are shown comparing control and Cdk1 knockdown wing disc cells. In the genome tracks shown in A, the "change" tracks are somewhat deceptive, as they are not displayed at the same scale as the control and knockdown tracks above them. This makes it seem that the log2 fold changes are more dramatic than they are. As an alternative, the tracks could be overlaid to show the difference. It was also unclear whether analyses were conducted to determine which changes across the genome were significantly different or if the analyses were purely qualitative.

2. More information about the H3K27ac/H3K27me/K9me peak distribution would have been valuable. Is there any relationship with the greatest changes and regions of chromatin accessibility? Are ATAC-seq data available from wildtype animals that can be used for this? The authors also mention ORC complex in the discussion. Are there any binding datasets that could be interrogated?

3. In Figure 4Q, a heatmap model is presented. Precisely what this means is unclear, particularly as not all methyl marks show changes.

---

## [Decision Letter · Decision Letter 2]

3 Mar 2026

Dear Dr Classen,

Thank you for your patience while we considered your revised manuscript "Cell cycle dynamics regulate H3K27 and H3K9 histone modifications" for publication as a Research Article at PLOS Biology. This revised version of your manuscript has been evaluated by the PLOS Biology editors, the Academic Editor and three of the original reviewers.

Based on the reviews, I am pleased to say that we are likely to accept this manuscript for publication, provided you satisfactorily address the remaining points raised by Reviewer #2. In addition, please make sure to address the following editorial and data-related requests that I have provided below (A-E):

(A) We routinely suggest changes to titles to ensure maximum accessibility for a broad, non-specialist readership. In this case, we would suggest a minor edit to the title, as follows. Please ensure you change both the manuscript file and the online submission system, as they need to match for final acceptance:

“Cell cycle dynamics regulate H3K27 and H3K9 histone modifications in Drosophila”

(B) Thank you for already providing the underlying data for the figures presented in the manuscript in the Source Data file. I have checked and this looks good, but it appears that Fig 5R and Fig 5R’ are mislabelled in the file (this should be Fig 5Q/5Q’?).

(C) Please ensure that each of the relevant figure legends in your manuscript include information on *WHERE THE UNDERLYING DATA CAN BE FOUND*, and ensure your supplemental data file/s has a legend.

(D) Please note that we cannot accept sole deposition of code in GitHub, as this could be changed after publication. However, you can archive this version of your publicly available GitHub code to Zenodo. Once you do this, it will generate a DOI number, which you will need to provide in the Data Accessibility Statement (you are welcome to also provide the GitHub access information). See the process for doing this here: https://docs.github.com/en/repositories/archiving-a-github-repository/referencing-and-citing-content

(E) Please ensure that your Data Statement in the submission system accurately describes where your data can be found and is in final format, as it will be published as written there.

We expect to receive your revised manuscript within two weeks.

*Published Peer Review History*

*Press*

Best wishes,

Richard

Richard Hodge, PhD

rhodge@plos.org

Reviewer remarks:

Reviewer #1: The authors have addressed all my concerns. I am pleased to recommend publication of the manuscript.

Reviewer #2: The authors have addressed all of my comments, and I think it is fine not to include the analyses that I asked for in order to keep the story focused. However, the quantification in Figure 5O'-Q' is still confusing to me, because it looks as if there is less H3K27ac in early S-phase upon CBP overexpression. If I understand the response to Reviewer 1 correctly, H3K27ac intensities are not absolute values and were not normalized to G1, but to the mean H3K27ac signal across all FUCCI-defined phases (i.e., the global average). This should be made much more clear in the text and Figure legend.

Reviewer #4: The authors have addressed my concerns. Their resubmitted manuscript is greatly improved.

---

## [Editor Report · Decision Letter 3]

11 Mar 2026

Dear Dr Classen,

On behalf of my colleagues and the Academic Editor, Tom Misteli, I am pleased to say that we can accept your manuscript for publication, provided you address any remaining formatting and reporting issues. These will be detailed in an email you should receive within 2-3 business days from our colleagues in the journal operations team; no action is required from you until then. Please note that we will not be able to formally accept your manuscript and schedule it for publication until you have completed any requested changes.

PRESS

Best wishes,

Richard

Richard Hodge, PhD

rhodge@plos.org

PLOS
